# Robust effects of corticothalamic feedback and behavioral state on movie responses in mouse dLGN

**Martin A Spacek[1]\*, Davide Crombie[1,2†], Yannik Bauer[1,2†], Gregory Born[1,2†], Xinyu Liu[1,2], Steffen Katzner[1], Laura Busse[1,3]\***

[1]Division of Neurobiology, Faculty of Biology, LMU Munich, Planegg-Martinsried, Germany; [2]Graduate School of Systemic Neurosciences, LMU Munich, Munich, Germany; [3]Bernstein Centre for Computational Neuroscience, Munich, Germany

**Abstract** Neurons in the dorsolateral geniculate nucleus (dLGN) of the thalamus receive a substantial proportion of modulatory inputs from corticothalamic (CT) feedback and brain stem nuclei. Hypothesizing that these modulatory influences might be differentially engaged depending on the visual stimulus and behavioral state, we performed in vivo extracellular recordings from mouse dLGN while optogenetically suppressing CT feedback and monitoring behavioral state by locomotion and pupil dilation. For naturalistic movie clips, we found CT feedback to consistently increase dLGN response gain and promote tonic firing. In contrast, for gratings, CT feedback effects on firing rates were mixed. For both stimulus types, the neural signatures of CT feedback closely resembled those of behavioral state, yet effects of behavioral state on responses to movies persisted even when CT feedback was suppressed. We conclude that CT feedback modulates visual information on its way to cortex in a stimulus-dependent manner, but largely independently of behavioral state.

**\*For correspondence:**
m.spacek@lmu.de (MAS);
busse@bio.lmu.de (LB)

†These authors contributed equally to this work

**Competing interest:** The authors declare that no competing interests exist.

## Editor's evaluation

This paper will be of interest to neuroscientists interested in understanding the role of corticothalamic feedback in coding of sensory inputs. The authors show that feedback is stronger for natural stimuli compared to artificial stimuli. Surprisingly, the feedback from the cortex works in parallel with other modulatory influences reflecting changes in the arousal (measured here with pupil size) or changes in locomotion.

## Introduction

Mammalian vision is based on a hierarchy of processing stages that are connected by feedforward circuits projecting from lower to higher levels, and by feedback circuits projecting from higher to lower levels. Feedforward processing is thought to create feature selectivity (*Lien and Scanziani, 2018*; *Hubel and Wiesel, 1962*) and invariance to low-level stimulus features (*Hubel and Wiesel, 1962*; *Chance et al., 1999*; *Riesenhuber and Poggio, 1999*; *Riesenhuber and Poggio, 2000*), to ultimately enable object recognition (*DiCarlo et al., 2012*). Hypotheses about the functional role of feedback circuits include top-down attention, working memory, prediction, and awareness (*Squire et al., 2013*; *Roelfsema and de Lange, 2016*; *Bastos et al., 2012*; *Lamme and Roelfsema, 2000*; *Takahashi et al., 2016*; *Larkum, 2013*). Compared to theories of feedforward processing, however, there is little consensus on the specific function of feedback connections (*Heeger, 2017*; *Gilbert and Li, 2013*).

Feedback in the mammalian visual system targets brain areas as early as the dorsolateral genic-ulate nucleus (dLGN) of the thalamus, where up to 30% of synaptic connections onto relay cells are established by corticothalamic (CT) feedback (*Sherman and Guillery, 2002*). Direct CT feedback is thought to arise from V1 layer 6 (L6) CT pyramidal cells (*Briggs, 2010*; *Sillito and Jones, 2002*), which are known for their notoriously low firing rates (*Vélez-Fort et al., 2014*; *Stoelzel et al., 2017*; *Crandall et al., 2017*; *Oberlaender et al., 2012*; *Swadlow, 1989*; *Pauzin and Krieger, 2018*), their sharp tuning for orientation (*Vélez-Fort et al., 2014*; *Liang et al., 2021*), and their diverse signaling of behavioral state (*Augustinaite and Kuhn, 2020*; *Liang et al., 2021*). The action of CT feedback on dLGN activity is generally considered modulatory rather than driving (*Sherman and Guillery, 1998*), as CT feedback inputs contact the distal dendrites of relay cells via NMDA glutamate (*Augustinaite et al., 2014*) or mGluR1 metabotropic receptors (*Godwin et al., 1996*), implying rather slow and long-lasting effects on dLGN activity. Similar to other depolarizing inputs to dLGN, such as neuromod-ulatory brain stem inputs (*McCormick, 1992*), CT feedback has been linked to promoting switching from burst to tonic firing mode, and to facilitating transmission of retinal signals (*Augustinaite et al., 2014*; *de Labra et al., 2007*; *Wang et al., 2006*; *Dossi et al., 1992*). However, since L6 CT pyramidal cells provide both direct excitation and indirect inhibition of dLGN via the thalamic reticular nucleus (TRN) and dLGN inhibitory interneurons (*Sillito and Jones, 2002*; *Usrey and Sherman, 2019*), the effects of CT feedback are expected to be complex and dependent on temporal and spatial aspects of the stimulus (*Crandall et al., 2015*; *Born et al., 2021*; *Murphy and Sillito, 1987*; *McClurkin and Marrocco, 1984*; *Jones et al., 2012*; *Hasse and Briggs, 2017*).

Most of the previous in vivo studies have probed the functional role of CT feedback with artifi-cial stimuli, and often in anesthetized animals; CT feedback, however, might be most relevant for processing of dynamic naturalistic information and during wakefulness. From a conceptual perspec-tive, if the role of feedback was to provide context based on an internal model built from the statistics of the world (*Berkes et al., 2011*; *Lee and Mumford, 2003*; *Rao and Ballard, 1999*; *Clark, 2013*), natural stimuli would be expected to best comply with this model, and hence better drive these feed-back mechanisms. Indeed, it has previously been suggested that CT feedback might be more strongly engaged for moving compared to stationary stimuli (*Sillito and Jones, 2002*), and for complex dynamic noise textures than simple moving bars (*Gulyás et al., 1990*), consistent with a potential role in figure-ground processing (*Poltoratski et al., 2019*; *Sillito et al., 1993*; *Cudeiro and Sillito, 1996*). Furthermore, since the responsiveness of feedback projections (*Makino and Komiyama, 2015*; *Keller et al., 2020*), including those originating from V1 CT neurons (*Briggs and Usrey, 2011*), seem to be strongly reduced by anesthesia, it is critical to examine CT feedback effects in awake animals. Indeed, L6CT neurons have recently been found to have diverse response modulations according to pupil-indexed behavioral state (*Augustinaite and Kuhn, 2020*).

Here, we recorded spiking activity in dLGN of awake mice and investigated how CT feedback affected dLGN responses to naturalistic movie clips. Suppressing CT feedback either via photostim-ulation of V1 parvalbumin-positive (PV+) inhibitory interneurons or via direct photosuppression of L6CT neurons, we found that CT feedback had consistent modulatory effects on dLGN responses to movie clips, which could largely be captured by an increase in gain. Effects of CT feedback on dLGN responses to grating stimuli were more diverse, highlighting the stimulus-dependency of CT feedback effects. Finally, while geniculate responses to movies during V1 suppression resembled those during quiescence, we found effects of CT feedback and behavioral state to be largely independent. Overall, our results demonstrate that neural responses to naturalistic movies en route to cortex can be robustly modulated by extra-retinal influences such as cortical feedback and behavioral state, which seem to be largely conveyed via different modulatory pathways.

## Results

### CT feedback robustly modulates dLGN responses to naturalistic movie clips

To investigate the impact of CT feedback on visual processing of naturalistic stimuli, we presented to head-fixed mice full-screen movie clips and compared responses of dLGN neurons during optogenetic suppression of V1 activity to a control condition with CT feedback left intact (*Figure 1 and* -Supplement 1). The responses of individual dLGN neurons to naturalistic movie clips were characterized by distinct

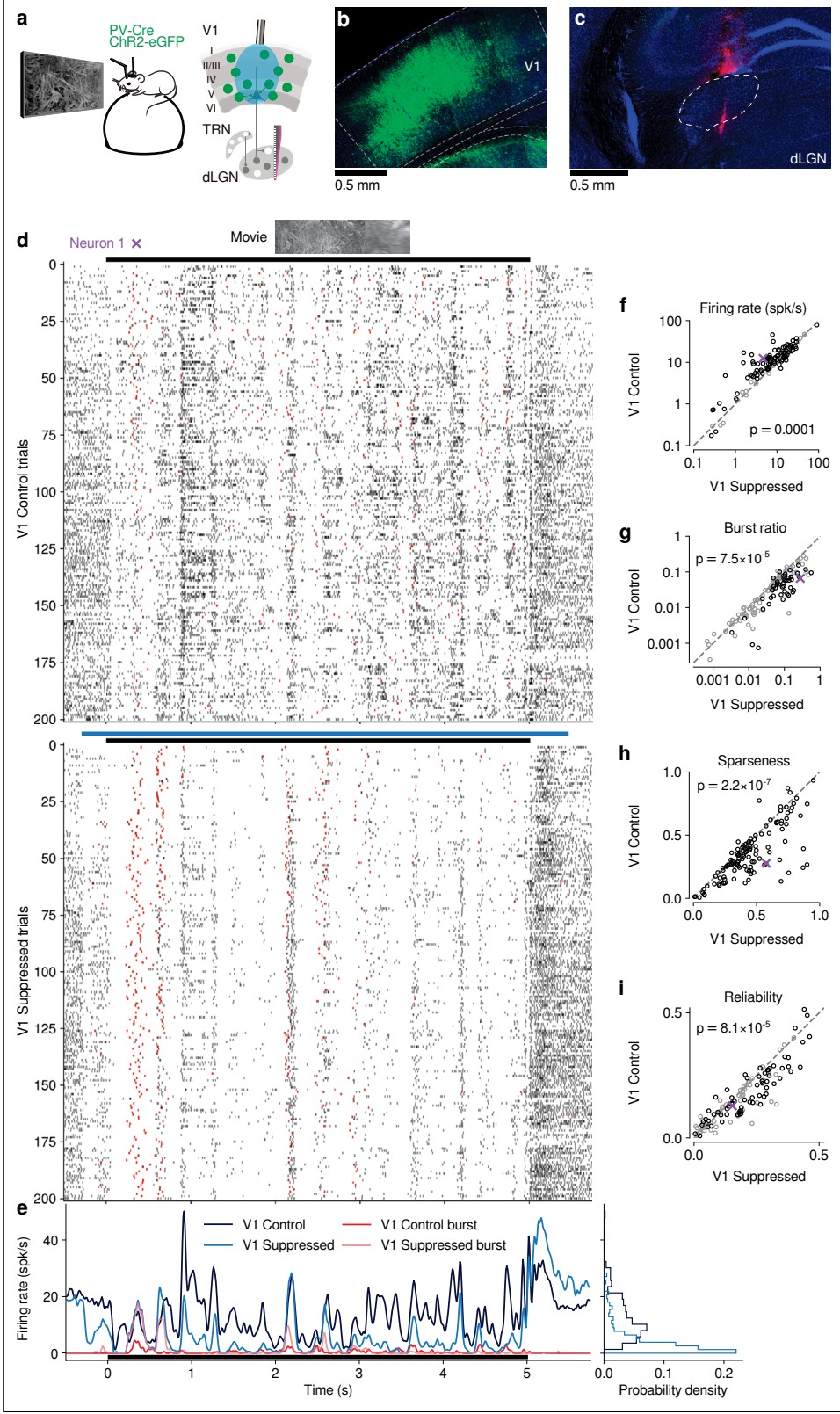

**Figure 1.** CT feedback modulates dLGN responses to full-screen naturalistic movie clips. (**a**) *Left*: Schematic of experimental setup. Head-fixed mice were placed on a floating Styrofoam ball and visual stimuli were presented on a screen located ~25 cm away from the animal. *Right*: ChR2 was conditionally expressed in PV + inhibitory interneurons (*green*) in all layers of V1 using a viral approach. Extracellular silicon electrode recordings were

*Figure 1 continued*

performed in dLGN with and without optogenetic suppression of V1. (**b**) Coronal section close to the V1 injection site for an example PV-Cre mouse (*blue*: DAPI; *green*: eYFP; Bregma: -3.4 mm). (**c**) Coronal section at the dLGN (white outline) recording site, same animal as in (**b**). For post-mortem confirmation of the electrode position, the back of the probe was stained with DiI (*magenta*) for one of the recording sessions (*blue*: DAPI; Bregma: -1.82 mm). (**d**) Raster plots of an example neuron for 200 presentations of a 5 s naturalistic movie clip, with CT feedback intact (control condition, *top*) and during V1 suppression (*bottom*). *Red*: burst spikes; *black bar*: movie clip presentation; *light blue bar*: V1 suppression. (**e**) *Left*: PSTHs for both the control (*dark blue*) and V1 suppression (*light blue*) conditions. Superimposed are PSTHs of burst spikes only, separately for control (*red*) and V1 suppression (*pink*) conditions. *Right*: Corresponding instantaneous firing rate distributions. (f–i) Comparison of control vs. V1 suppression conditions for mean firing rate (**f**), burst ratio (**g**), temporal sparseness (**h**), and response reliability (**i**), all calculated for the duration of the movie clip. Sparseness captures the activity fraction of a neuron, re-scaled between 0 and 1 (*Vinje and Gallant, 2000*). Response reliability is defined as the mean Pearson correlation of all single trial PSTH pairs (*Goard and Dan, 2009*). For sample sizes, see *Table 1*. *Purple*: example neuron. Black markers in (**f,g,i**) indicate neurons with individually significant effects (Welch's t-test). See also *Figure 1—figure supplement 1* to *Figure 1—figure supplement 6*.

The online version of this article includes the following video and figure supplement(s) for figure 1:

**Figure supplement 1.** Confirmation of optogenetic suppression of V1 responses and targeting dLGN for recordings.

**Figure supplement 2.** Effects of CT feedback on additional parameters of responses to naturalistic movies and their relationship with firing rate.

**Figure supplement 3.** Feedback effects during movie presentation are largely independent of functional cell type classification.

**Figure supplement 4.** Selective optogenetic suppression of L6 CT feedback in Ntsr1-Cre yielded similar results as global V1 suppression via PV + activation.

**Figure supplement 5.** Photostimulation in an Ntsr1- control mouse injected with cre-dependent stGtACR2 had no effect on neural responses.

**Figure supplement 6.** Effects of photostimulation on pupil size were unrelated to CT feedback effects on dLGN neuronal activity.

**Figure 1—video 1.** First example 5 s movie clip used for visual stimulation.

https://elifesciences.org/articles/70469/figures#fig1video1

**Figure 1—video 2.** Second example 5 s movie clip used for visual stimulation.

https://elifesciences.org/articles/70469/figures#fig1video2

response events that were narrow in time and reliable across trials (*Figure 1d*, top, example neuron). Consistent with the notion that CT feedback has a modulatory rather than driving role (*Sherman, 2016*), even during V1 suppression this temporal response pattern remained somewhat preserved (Pearson correlation $r = 0.54$, $p < 10^{-6}$, *Figure 1d and e*). Yet, as illustrated in the example neuron, with CT feedback intact, firing rates were higher and burst spikes were less frequent (*Figure 1e*, left). Accordingly, the distributions of instantaneous firing rates in the two conditions were significantly different (KS test, $p < 10^{-6}$), and were more skewed during V1 suppression than with CT feedback intact ($\gamma = 2.02$ vs 1.22; *Figure 1e*, right).

We observed similar effects in the recorded population of dLGN neurons, where CT feedback enhanced overall responses and promoted tonic firing mode. Indeed, while mean firing rates varied almost 4 orders of magnitude across the population (~ 0.1–100 spikes/s), they were higher in control conditions with CT feedback intact than during V1 suppression (13.7 vs 10.5 spikes/s; linear multilevel-model (LMM): $F_{1,63.2} = 17.1$, $p = 0.0001$; *Figure 1f*). In addition, CT feedback also influenced more fine-grained properties of geniculate responses. First, with CT feedback, the mean proportion of spikes occurring as part of a burst event was about half of what we observed during suppression (0.05 vs 0.09; LMM: $F_{1,64.0} = 17.9$, $p = 7.5 \times 10^{-5}$; *Figure 1g*). Second, consistent with the distributions of firing rate for the example neuron (*Figure 1e*, right), responses to the naturalistic movie clips with CT feedback intact were, on average, less sparse (0.35 vs 0.45; LMM: $F_{1,63.0} = 33.7$, $p = 2.2 \times 10^{-7}$; *Figure 1h*), indicating that neurons fired less selectively across the frames of the movie. Finally, we also examined the effect of CT feedback on response reliability. To quantify reliability, we computed the Pearson correlation coefficient of a neuron's responses between each pair of the 200 stimulus repeats

per condition, and averaged the correlation coefficients over all pair-wise combinations (*Goard and Dan, 2009*). With CT feedback intact, mean response reliability was lower than without feedback (0.15 vs 0.18; LMM: $F_{1,63.1} = 17.8, p = 8.1 \times 10^{-5}$; *Figure 1i*). Except for the effects on sparseness, the feedback effects on responses to naturalistic movies were unrelated to changes in firing rates (*Figure 1—figure supplement 2c-g*). The increased trial-to-trial reliability during V1 suppression could not be explained by higher stability in eye positions, because variability in eye position was slightly larger with CT feedback intact vs. suppressed (*Figure 1—figure supplement 2h*), and effects of CT feedback on neural reliability were unrelated to changes in variability of eye position (*Figure 1—figure supplement 2i*). Splitting the dLGN population into putative cell types according to several functional characteristics and location within dLGN revealed few differences in how global V1 suppression affected firing rates and bursting (*Figure 1—figure supplement 3*). As V1 suppression by PV +activation is robust, yet lacks selectivity (*Wiegert et al., 2017*), we repeated our experiments while directly photo-suppressing L6CT neurons. To this end, we expressed the inhibitory opsin stGtACR2 (*Mahn et al., 2018*) in V1 Ntsr1+ neurons, which correspond to $\geq 90\%$ to L6 CT neurons (*Bortone et al., 2014*; *Kim et al., 2014*, *Figure 1—figure supplement 4*). These experiments with specific suppression of L6 CT neurons during viewing of naturalistic movies yielded identical conclusions (*Figure 1—figure supplement 4a-h*).

Lastly, we performed two additional controls to rule out that photostimulation *per se* caused our findings. First, we repeated our experiments on an Ntsr1- control mouse, which was injected and underwent the same visual and photostimulation protocol. This negative control mouse did not show any effects of photostimulation on dLGN responses (*Figure 1—figure supplement 5a-d*). Second, we identified those experiments (14/31 for PV + activation, 0/10 for Ntsr1 + suppression experiments), where photostimulation decreased pupil size, indicative of light leakage into the retina. Even with these sessions removed, we found that our results remained qualitatively unchanged (*Figure 1—figure supplement 6a-f*). Finally, considering again all recordings, the effects of CT feedback on neuronal activity were unrelated to light-induced changes in pupil size (*Figure 1—figure supplement 6g-j*). Together, these results rule out that photostimulation *per se* led to the modulation of dLGN responses during naturalistic movie viewing.

Taken together, our results indicate that CT feedback can robustly modulate responses of dLGN neurons to naturalistic movie clips. The modulations are consistent with a net depolarizing effect, which supports higher firing rates and more linear, tonic firing mode with higher dynamic range, at the expense of sparseness, trial-to-trial reliability, and signal-to-noise.

## V1 suppression decreases dLGN responses to naturalistic movies by reducing response gain

To better understand the effects of V1 suppression on dLGN firing rate, we next asked whether the observed reduction in responsiveness could be explained by a divisive and/or subtractive change (*Figure 2*). Using repeated random subsampling cross-validation, we fit a simple threshold linear model (*Figure 2a, inset*) to timepoint-by-timepoint responses in suppression vs. feedback conditions, and extracted the slope and threshold of the fit for each subsample (*Figure 2b and d*). In the two example neurons shown in *Figure 2a–d*, the fitted slope was significantly smaller than 1 (neuron 2: median slope of 0.66, 95% CI: 0.63–0.69, *Figure 2b*; neuron 1: median slope of 0.37, 95% CI: 0.32–0.41, *Figure 2d*), while the threshold ($x$-intercept) was either small or not significantly different from 0 (neuron 2: median of 1.58, 95% CI: 0.39–2.91; neuron 1: median of $-0.14$, 95% CI: $-1.49$–0.89). We obtained similar results for the population of recorded neurons, where V1 suppression decreased the neurons' responses to naturalistic movie clips via a substantial change in response gain (slope of $0.75 \pm 0.1$; LMM) without a significant shift in baseline (threshold of $-0.19 \pm 1.15$; LMM; *Figure 2e*). This demonstrates that V1 suppression influences responses in dLGN to naturalistic movie clips predominantly via a divisive effect.

We noticed that the threshold linear model could predict the effects of V1 suppression better for some neurons than for others. We therefore explored whether poor fits of the model might be related to our finding that V1 suppression can trigger non-linear, burst-mode firing. For instance, the threshold-linear model accurately captured the responses of example neuron 2 (median $R^2 = 0.90$, cross-validated; *Figure 2a and b*), which exhibited little bursting during V1 suppression (burst ratio: 0.007). Neuron 1, in contrast, had a higher burst ratio during suppression (0.28) and the prediction

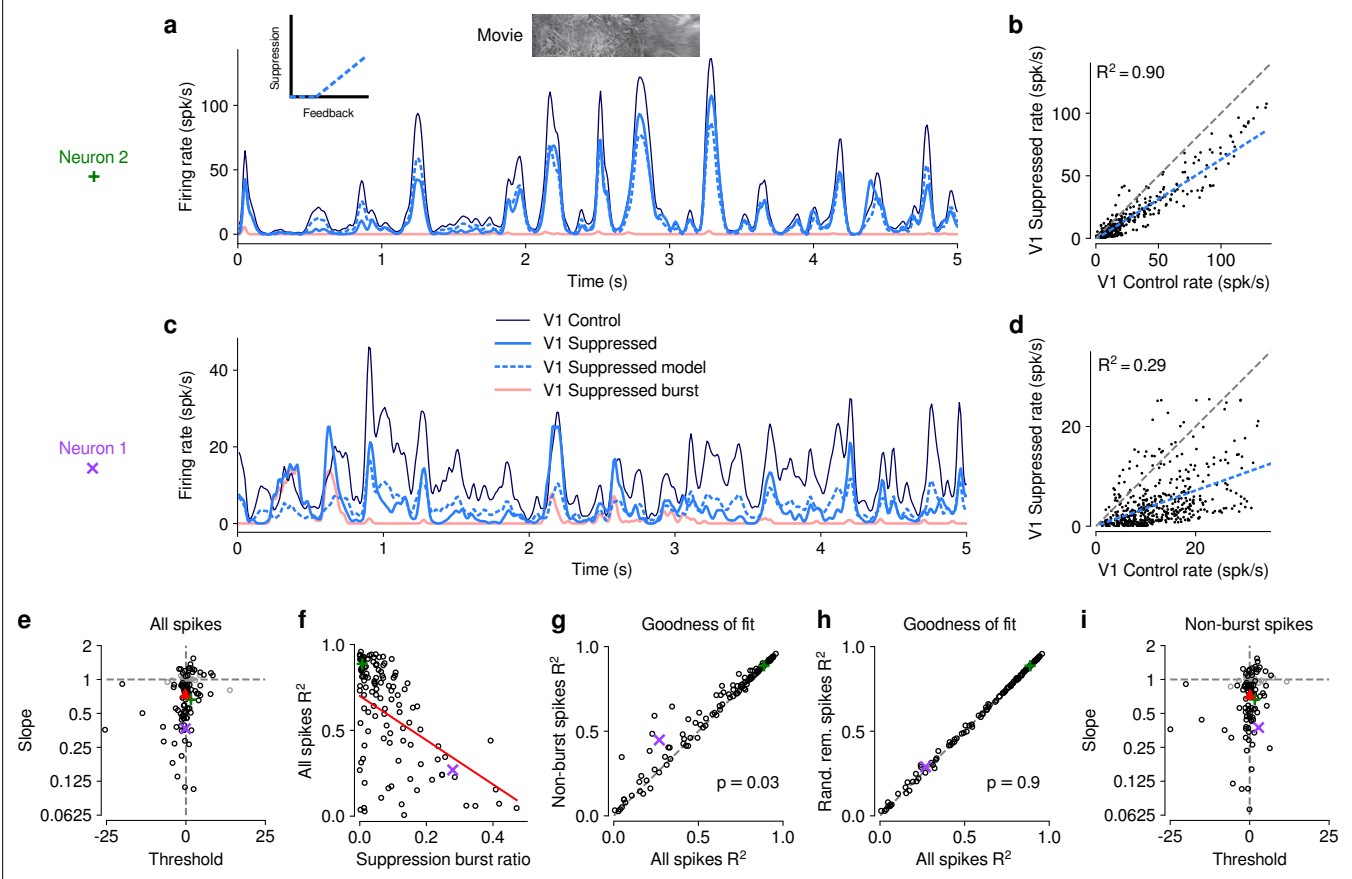

**Figure 2.** The effect of V1 suppression on dLGN responses to naturalistic movie clips is predominantly divisive. (**a**) PSTHs of an example neuron during control (*dark blue*) and V1 suppression (*light blue*) conditions, for a random subset of 50% of trials per condition not used for model fitting. Responses during the V1 suppression condition are approximated by the threshold linear model (*dashed light blue*) based on responses during the control condition. *Pink:* PSTH during V1 suppression for burst spikes only. *Inset:* cartoon of threshold linear model. (**b**) Timepoint-by-timepoint comparison of instantaneous firing rates of the PSTHs (derived from the 50% of trials not used for fitting) during the suppression vs. feedback conditions. PSTH data points are plotted at 0.01ms resolution. *Dashed light blue line:* threshold linear model fit. (**c,d**) Same as (**a,b**) for a second example neuron (same as in *Figure 1d and e*). (**a,b**) and (**c,d**) each contain data from 1 representative subsample. (**e**) Slope and threshold parameters for all neurons. Each point represents the median for each neuron across 1000 random subsamples of trials. Black points indicate neurons with slopes significantly different from 1 (95% CI). (**f**) Cross-validated model prediction quality (median $R^2$) vs. burst ratio during V1 suppression. *Red line:* LMM fit. (**g**) Model prediction quality $R^2$ with and without removal of burst spikes. (**h**) Model prediction quality with and without removal of an equivalent number of tonic spikes. (**i**) Same as (**e**) but with burst spikes removed. (**e–h**) *Purple, green:* example neurons; *red triangle:* LMM estimate of the mean.

sometimes overestimated or underestimated peaks in the actual response, such that the percentage of explained variability was rather low (median $R^2 = 0.29$, cross-validated, *Figure 2c and d*).

Indeed, across the population of recorded neurons, the model goodness of fit (median $R^2$, cross-validated) during V1 suppression was inversely related to the burst ratio (slope of $-1.29 \pm 0.5$; LMM; *Figure 2f*), consistent with the notion that the highly non-linear, all-or-none-like burst mode firing (*Sherman, 2001*) cannot be captured by the threshold-linear model (see also *Lesica and Stanley, 2004*). To further investigate the impact of bursting on response transformations by CT feedback, we re-computed the PSTHs for each neuron during V1 suppression after removing all burst spikes. Removal of burst spikes allowed our model to capture the effects of V1 suppression even better (all spikes: mean $R^2 = 0.58$; non-burst spikes: mean $R^2 = 0.61$; LMM; $F_{1,160.8} = 4.8$, $p = 0.03$; *Figure 2g*). Importantly, this increase in model performance was not simply a consequence of removing a certain proportion of spikes that originally needed to be predicted: discarding an equivalent number of randomly selected tonic spikes did not yield improved fit quality (random tonic spikes removed: mean $R^2 = 0.58$; LMM; $F_{1,162} = 0.005$, $p = 0.9$; *Figure 2h*). While burst spikes cannot be captured by the threshold-linear model, removing burst spikes, however, did not change our conclusion that the

effect of V1 suppression on movie responses was predominantly divisive (slope: $0.74 \pm 0.09$; threshold: $0.09 \pm 1.3$; LMM; *Figure 2i*), likely because burst events were much rarer than tonic spikes (see also *Figure 1g*). Indeed, firing mode (all spikes vs. non-burst spikes) had no effect on either slope (LMM: $F_{1,162.7} = 0.6$, $p = 0.4$) or threshold estimates (LMM: $F_{1,157.3} = 0.2$, $p = 0.7$) of the simple linear model. Together, these results show that V1 suppression decreases dLGN responses to naturalistic movies mostly by reducing response gain.

## CT feedback modulates dLGN responses evoked by drifting gratings

Previous studies have investigated the effects of CT feedback using artificial stimuli, such as gratings and bars (*Olsen et al., 2012*; *Denman and Contreras, 2015*; *Wang et al., 2006*; *Murphy and Sillito, 1987*). To relate our findings to these studies, and to investigate the role of stimulus type, we next examined the effects of V1 suppression during the presentation of drifting gratings (*Figure 3*). To approximate the visual stimulus configuration used for naturalistic movie clips, we presented full-screen gratings drifting in one of 12 different orientations, and selected a pseudo-random subset of trials for V1 suppression. As expected, we found that many single dLGN neurons in the control condition with CT feedback responded at the temporal frequency (TF, 4 cyc/s) of the drifting grating (*Figure 3a and b*). Similar to previous studies in mouse dLGN (*Piscopo et al., 2013*; *Román Rosón et al., 2019*; *Marshel et al., 2012*), we also encountered some dLGN neurons with tuning for grating orientation or direction (*Figure 3*, a2, b).

Contrary to the robust effects of CT feedback on movie responses, V1 suppression had mixed effects on dLGN responses to drifting gratings. Example neuron 1, for instance, had lower firing rates with CT feedback intact, both in the orientation tuning (*Figure 3*, a₂) and the cycle-averaged response to the preferred orientation (*Figure 3a*3). In addition, in control conditions with CT feedback intact, there were markedly fewer burst spikes. In contrast, example neuron 3 responded more strongly with CT feedback intact (*Figure 3*, b₂,₃). Such diverse effects of CT feedback, as reported before for anesthetized mice (*Denman and Contreras, 2015*), were representative of the recorded population (*Figure 3c*): V1 suppression during grating presentation significantly reduced responses for some neurons, but significantly increased responses for others, such that the average firing rates in the two conditions were almost identical (control: 14.5 spikes/s, V1 suppression: 15.0 spikes/s) and statistically indistinguishable (LMM: $F_{1,43.0} = 0.15$, $p = 0.70$). In contrast to these diverse effects on firing rate, but similar to our findings for naturalistic movie clips, intact CT feedback was consistently associated with less bursting (burst ratios of 0.043 vs 0.15; LMM: $F_{1,43.0} = 25.3$, $p = 9.2 \times 10^{-6}$; *Figure 3d*). Also similar to our findings for movies, there was no relationship between the strength of feedback effects on firing rate and on bursting (LMM: slope $0.029 \pm 0.41$, *Figure 4—figure supplement 1a*).

Beyond studying overall changes in responsiveness and firing mode, we next asked how CT feedback affected the tuning for grating orientation of dLGN neurons. It is known from previous studies (*Piscopo et al., 2013*; *Cruz-Martín et al., 2014*; *Marshel et al., 2012*; *Zhao et al., 2013*; *Scholl et al., 2013*) that mouse dLGN neurons show various degrees of orientation tuning, ranging from few strongly tuned neurons, potentially relaying tuned input from the retina (*Cruz-Martín et al., 2014*), to a larger group with orientation bias (*Piscopo et al., 2013*; *Scholl et al., 2013*). We computed orientation tuning curves separately for control conditions with CT feedback and V1 suppression conditions. For neuron 1, intact CT feedback was associated not only with lower average firing rates, but also poorer selectivity (OSIs of 0.14 vs 0.25; *Figure 3*, a₂). In contrast, for neuron 3, orientation selectivity was similar during control and V1 suppression conditions (OSIs of 0.1 vs 0.09; *Figure 3*, b₂). These results were representative of the population, where CT feedback affected orientation selectivity in diverse ways, with virtually no difference in population means (control OSI: 0.13; V1 suppression: 0.12; LMM: $F_{1,88.7} = 0.31$, $p = 0.58$; *Figure 3e*; see also *Scholl et al., 2013*; *Li et al., 2013*; *Lien and Scanziani, 2013*; *Denman and Contreras, 2015*). For neurons with OSI > 0.02 and well-fit orientation tuning curves ($R^2 > 0.5$), preferred orientation during feedback and suppression conditions was largely similar, except for some cases where it shifted (*Figure 3f*). As was the case for movies, splitting the dLGN population into putative cell types according to several functional characteristics and their location within dLGN revealed few consistent differences in how global V1 suppression during gratings affected firing rates and bursting (*Figure 3—figure supplement 1*). Taken together, although effects of V1 suppression on firing rate were more diverse in magnitude and sign for grating stimuli, the similarity of orientation

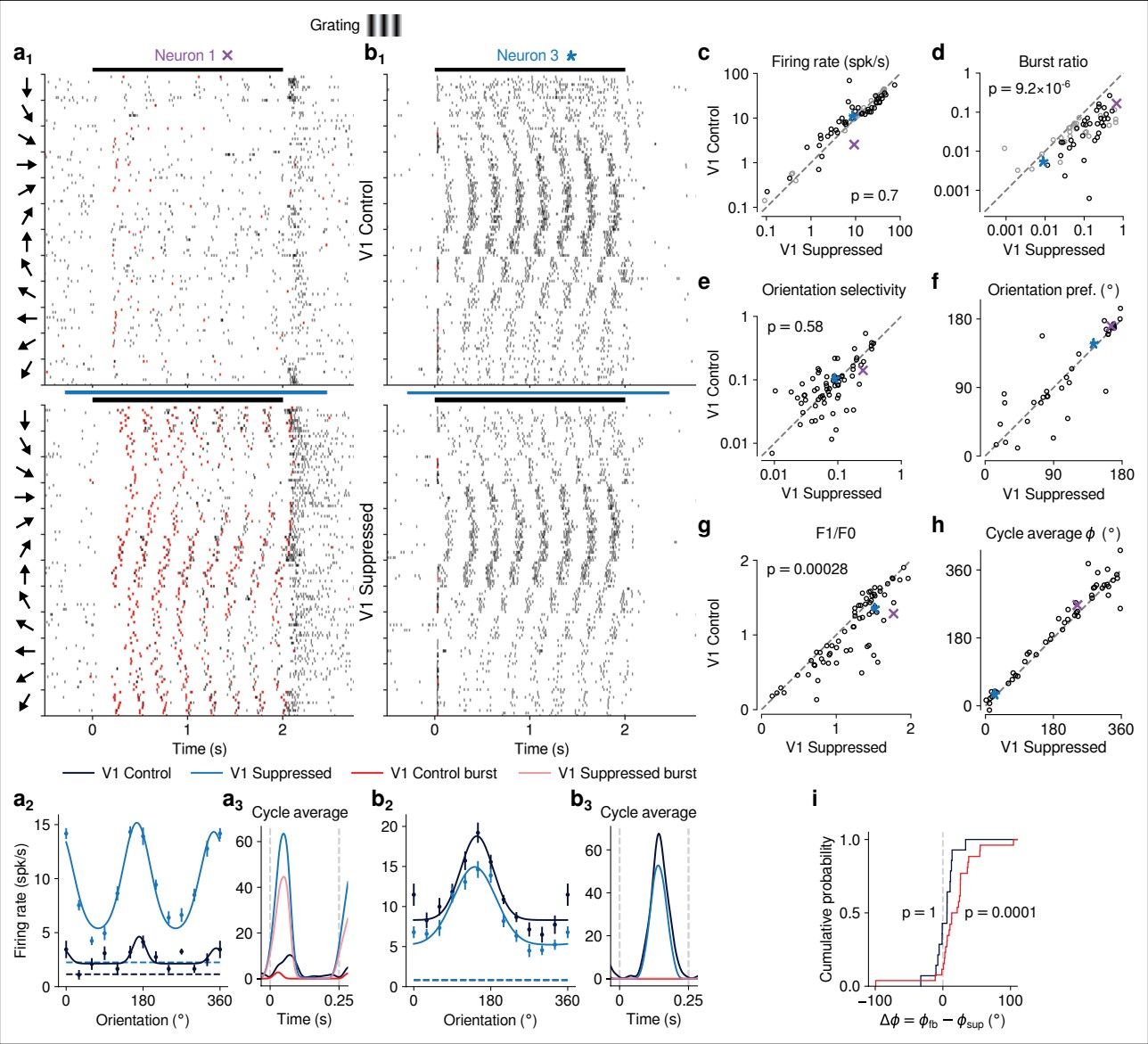

**Figure 3.** CT feedback modulates dLGN responses to drifting gratings. (**a**) Responses of example neuron 1 (same as in *Figures 1d, e ,, 2c and d*) to full-screen, drifting gratings. (**a**1) Raster plot in response to drifting gratings, with trials sorted by grating orientation (10 trials per orientation, 30° steps). *Red*: burst spikes; *black bar*: grating stimulation; *light blue bar*: V1 suppression. (**a**2) Corresponding orientation tuning curve. Dashed lines represent spontaneous firing rates in response to medium gray screen. *Error bars*: standard error of the mean. (**a**3) Cycle average response to preferred orientation. *Dark blue, light blue*: cycle average constructed from all spikes. *Red, pink*: cycle average constructed from burst spikes only. *Dark blue, red*: Control condition with CT feedback intact; *light blue, pink*: V1 suppression. (**b**) Same as (**a**), for another example neuron (example neuron 3). (c–h) Comparison of the control conditio with CT feedback intact vs. V1 suppression, for mean firing rate (**c**), burst ratio (**d**), orientation selectivity index (OSI) (**e**), preferred orientation $\theta$ (**f**), $F_1/F_0$ (**g**), and cycle average phase $\phi$ (**h**). *Purple, blue*: example neurons. Black markers in (**c,d**) indicate neurons with individually significant effects (Welch's t-test). (**i**) Cumulative distribution of cycle average phase differences between control and V1 suppression conditions. *Dark blue*: neurons with little burst spiking (ratio of cycle average peak for burst spikes to cycle average peak for all spikes < 0.1); *red*: neurons with substantial burst spiking (ratio of cycle average peak for burst spikes to cycle average peak for all spikes ≥ 0.1).

The online version of this article includes the following figure supplement(s) for figure 3:

**Figure supplement 1.** As for movies (*Figure 1—figure supplement 3*), feedback effects during grating presentation are largely independent of functional cell type classification.

selectivity between CT feedback conditions suggests underlying changes in gain, in accordance with what we observed for naturalistic movies.

Inspecting the spike rasters at different orientations, we realized that dLGN neurons appeared to have a stronger response component at the grating's temporal frequency during V1 suppression than when feedback was intact (*Figure 3*, $a_1$). To test whether V1 suppression affected the ability of dLGN to respond at the gratings' temporal frequency, for each neuron we computed the amplitude of the response at the stimulus frequency ($F_1$ component) relative to the mean response ($F_0$ component) (*Skottun et al., 1991*; *Carandini et al., 1997*) and found that $F_1/F_0$ ratios were indeed lower when feedback was intact (1.08 vs 1.22; LMM: $F_{1,43.5} = 15.6$, $p = 0.00028$; *Figure 3g*). To explore the impact of CT feedback on the F1 response component in more detail, we examined the cycle average responses to the preferred orientation, and asked how CT feedback affected response phase. Similar to the results obtained for the example neurons (*Figure 3*, $a_3$, $b_3$), we found that V1 suppression could advance response phase (*Figure 3h*). This phase advance occurred more often for neurons whose responses during V1 suppression included a substantial proportion of burst spikes (*Figure 3i*, *red*; 25 of 29 neurons showed phase advance, $p = 0.0001$, binomial test) than for neurons which during V1 suppression burst little or not all (*Figure 3i*, *dark blue*; 11 of 21 neurons advanced, $p = 1$, binomial test). In agreement with earlier findings from intracellular recordings in anesthetized cats (*Lu et al., 1992*), these analyses demonstrate that the phase advance is driven by the dynamics of burst spiking. Finally, as for our re-assessment of CT feedback effect on responses to naturalistic movies, our conclusions regarding the effects of CT feedback on grating responses did not change when we repeated our experiments using a selective suppression of Ntsr1 + neurons with stGtACR2 (*Mahn et al., 2018*, *Figure 1—figure supplement 4i-o*). Also, during grating experiments, the Ntsr1- mouse controlling for effects of photostimulation per se showed no effects on neural responses to gratings (*Figure 1— figure supplement 5e-i*).

## Effects of CT feedback on dLGN firing rates are more consistent and stronger overall for full-screen movies than full-screen gratings

Our analyses suggest that the impact of CT feedback on firing rates might be stronger overall for naturalistic movie stimuli than for gratings. To test this hypothesis, we focused on the subset of neurons recorded with both types of stimuli. Indeed, when we compared feedback modulation indices (FMIs, i.e. the difference between feedback conditions over their sum of firing rates), we found that FMI was

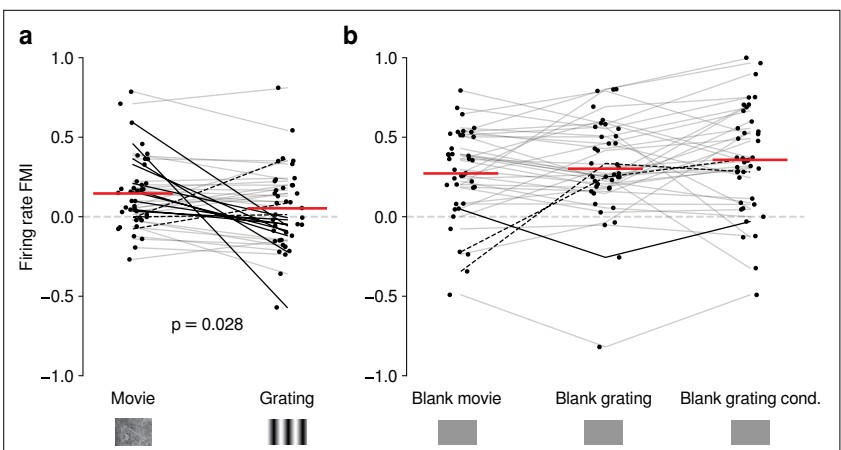

**Figure 4.** Effects of CT feedback on dLGN firing rate depend on stimulus type. (**a**) Comparison of the strength of CT feedback effects on firing rate (feedback modulation index, FMI) during presentation of full-screen movie clips and gratings. (**b**) Comparison of the strength of CT feedback effect on firing rate for blank stimuli interleaved with movies or gratings. *Red*: mean (LMM), *dark lines*: changes in sign of feedback modulation effect with stimulus type from positive for movies to negative for gratings (*solid*) and vice versa (*dashed*). For (**a**) and (**b**), we randomly jittered the horizontal position of the points to avoid overlap; lines connecting the paired samples still end at the central position to represent change. See also *Figure 4—figure supplement 1*.

The online version of this article includes the following figure supplement(s) for figure 4:

**Figure supplement 1.** Control analyses assessing the difference in CT feedback effects for gratings and movies.

on average more positive for movies than for gratings (0.15 vs 0.053; LMM: $F_{1,38} = 5.21$, $p = 0.028$; *Figure 4a*). Remarkably, in 10/39 neurons (*Figure 4a*, dark lines) V1 suppression decreased firing rates for movies (positive movie FMI), but increased firing rates for gratings (negative grating FMI). The opposite effect only occurred in 3/39 neurons (dark dashed lines). These findings were not a consequence of differences in firing rates that might have already been present in control conditions with CT feedback intact (*Figure 4—figure supplement 1b*), and were also not a consequence of the longer duration of V1 suppression during movie clips (*Figure 4—figure supplement 1c, d*).

The differences in the effects of CT feedback on firing rates during full-screen gratings vs. movies might be related to feedback-induced changes in bursting, which might be stimulus-dependent (*Lu et al., 1992*; *Grubb and Thompson, 2005*) and can drive high-frequency firing. To test this hypothesis, we compared CT feedback modulation of burst ratio for gratings vs. movie clips, and found that V1 suppression indeed induced stronger bursting for gratings than for movies (*Figure 4—figure supplement 1e*). However, for both movies (*Figure 1—figure supplement 2c*) and gratings (*Figure 4—figure supplement 1a*), CT feedback effects on firing rates were unrelated to those on bursting. Thus, while suppression of CT feedback engages bursting overall more strongly for gratings than movies, this differential recruitment does not seem to account for differences in CT feedback-related modulations of firing rates for movies vs. grating stimuli.

Differences in CT feedback effects between firing rates to full-screen gratings and movies might instead be related to differences in longer-lasting, systematic changes in neural activity, which might occur due to differential adaptation or differences in behavioral state induced by the two stimulus types. To address this possibility, we focused on periods of blank screen, which were contained in both stimulus types. These were short (~0.3 s) periods directly preceding each full-screen movie and grating trial (see e.g., *Figures 1d and 3a*), as well as blank trials interleaved as one condition in the grating experiments. Applying our analyses to these various blank stimuli (*Figure 4b*, *Figure 4—figure supplement 1g-i*), we found that CT feedback enhanced mean firing rates regardless of blank type or blank period duration (positive firing rate FMIs, mean FMIs: 0.27 vs. 0.30 vs. 0.36; LMM: $F_{2,76} = 1.69$, $p = 0.19$; *Figure 4b*). This CT feedback-related average enhancement for blank stimuli was even stronger than the enhancement observed during movie presentation (LMM: $F_{1,116} = 15.1$, $p = 0.0002$), and stronger than the mixed effects during grating presentation (LMM: $F_{1,116} = 34.9$, $p = 3.6 \times 10^{-8}$). Since the CT feedback effects on these various blank stimuli did not depend on blank period duration or whether blanks were embedded in grating or movie experiments (see also *Figure 4—figure supplement 1f-l*), we conclude that differences in longer lasting changes in neural activity or behavioral state did not underlie the differential effect of CT feedback for full screen movies vs. gratings. Instead, we interpret these findings to highlight that CT feedback modulates dLGN responses in a stimulus-dependent way. In particular, the strength and sign of CT feedback gain might be sensitive to features of the visual stimulus, such as the contrast, the dynamics, or the statistics of the center and the surround stimulation.

## Effects of behavioral state on dLGN responses resemble effects of CT feedback, but are largely independent

Previous studies have reported that responses of mouse dLGN neurons to grating stimuli are modulated by behavioral state as inferred by locomotion (*Erisken et al., 2014*; *Aydın et al., 2018*; *Williamson et al., 2015*). To assess how these findings extend to more complex stimuli, we separated the trials with CT feedback intact according to the animals' locomotion behavior. We considered trials as 'run trials' if the animal's speed exceeded 1 cm/s for at least 50% of the stimulus presentation and as 'sit trials' if the animal's speed fell below 0.25 cm/s for at least 50% of the stimulus presentation. When we examined the spike rasters and PSTHs of example neuron 1 in control conditions with CT feedback intact (*Figure 5a and b*), we found that, despite preserved temporal features of the responses (Pearson correlation $r = 0.72$ between run and sit PSTHs, $p < 10^{-6}$), firing rates were higher overall during locomotion than stationary periods. Additionally, during locomotion, the distribution of firing rates was less skewed ($\gamma = 1.15$ vs $1.45$ during stationary trials), with a decrease of low and an increase of medium firing rates (KS test, $p < 10^{-6}$). This pattern was also observed in the population of dLGN neurons, where firing rates were consistently higher for trials with locomotion compared to trials when the animal was stationary (11.9 vs 8.9 spikes/s; LMM: $F_{1,63.9} = 94.1$, $p = 3.5 \times 10^{-14}$; *Figure 5c*). Similar to previous reports using gratings (*Niell and Stryker, 2010*; *Erisken et al., 2014*), we found that

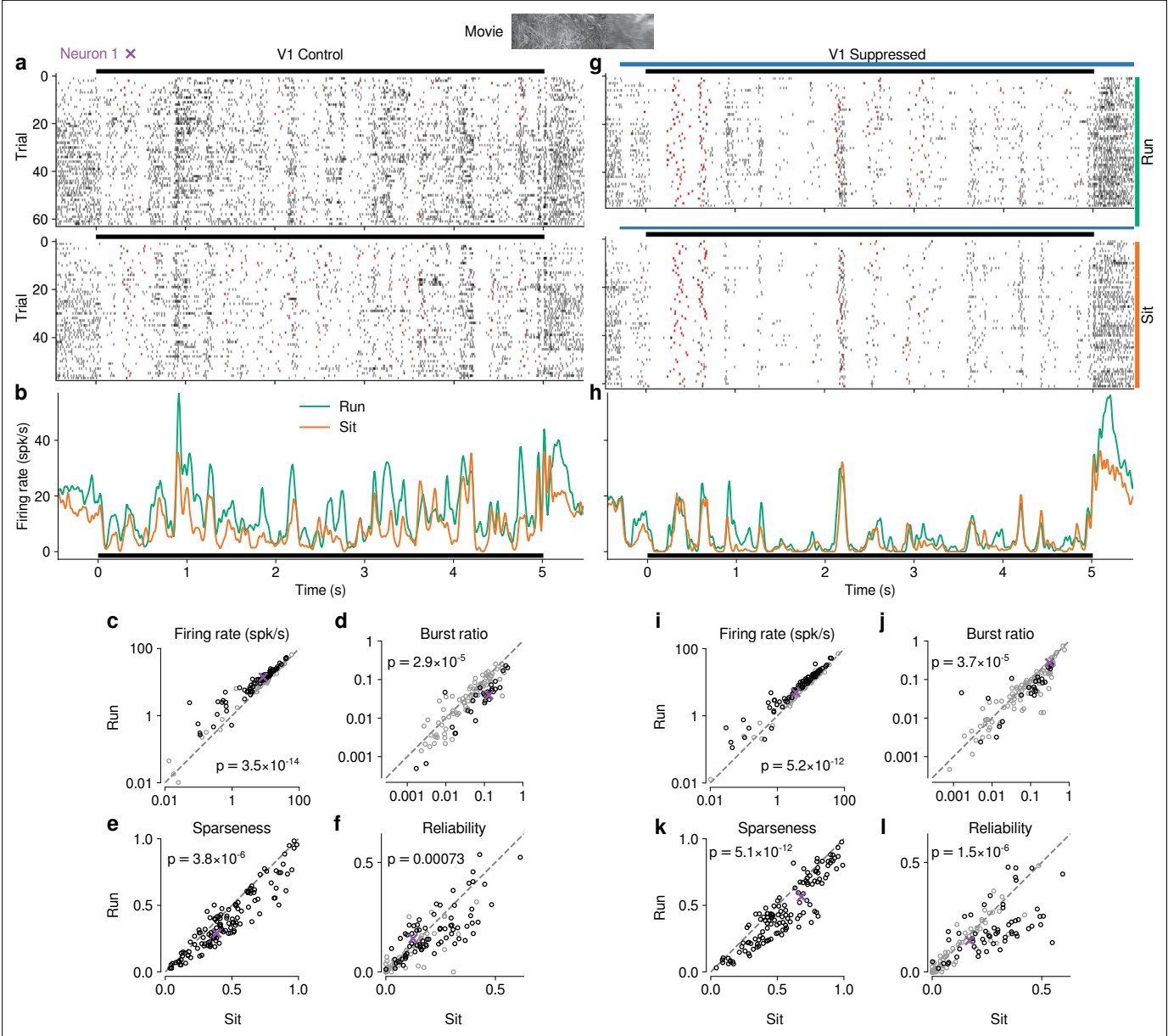

**Figure 5.** Effects of locomotion on dLGN responses resemble those of CT feedback, but persist even during V1 suppression. (**a**) Spike raster of example neuron 1 (same as *Figure 1d*) in response to a naturalistic movie clip during locomotion and stationary trials with CT feedback intact. *Top*: trials with run speed > 1 cm/s; *bottom*: trials with run speed <0.25 cm/s, both for at least > 50% of each trial. *Red*: burst spikes. (**b**) Corresponding PSTHs. *Green*: locomotion, *orange*: stationary; *black bar*: duration of movie clip. (**c–f**) Comparison of firing rates (**c**), burst ratio (**d**), sparseness (**e**), and trial-to-trial reliability (**f**) during locomotion and stationary trials. Black markers in (**c,d,f**) correspond to individually significant observations (Welch's t-test). (**g–l**) Same as (**a–f**), for locomotion and stationary trials during V1 suppression. *Light blue bar*: V1 suppression. See also *Figure 5—figure supplement 1*.

The online version of this article includes the following figure supplement(s) for figure 5:

**Figure supplement 1.** Effects of locomotion on additional parameters of responses to naturalistic movie clips and relationship with firing rate.

**Figure supplement 2.** Effects of pupil-indexed arousal on dLGN responses to movies.

bursting was lower during locomotion than stationary periods (0.035 vs 0.063; LMM: $F_{1,66.7} = 20.2$, $p = 2.9 \times 10^{-5}$; *Figure 5d*). Beyond these established measures, using movie clips allowed us to test the effects of locomotion on additional response properties: trials with locomotion were associated with lower sparseness (0.40 vs 0.47; LMM: $F_{1,181.9} = 22.8$, $p = 3.8 \times 10^{-6}$; *Figure 5e*) and lower trial-to-trial reliability (0.13 vs 0.16; LMM: $F_{1,176.1} = 11.8$; *Figure 5f*). This locomotion-related decrease of reliability could be related to, but is likely not fully explained by, the increase in eye movements typically

associated with running (*Figure 5—figure supplement 1h, i*; *Erisken et al., 2014*; *Bennett et al., 2013*). These analyses demonstrate that in dLGN, processing of naturalistic movie clips is robustly modulated by locomotion. Curiously, in all aspects tested, these modulations by locomotion had the same signatures as those of CT feedback: increased firing rates, reduced bursting, and decreased sparseness and trial-to-trial reliability.

Since the effects of CT feedback and locomotion closely resembled each other, and since L6CT neurons themselves are modulated by locomotion (*Augustinaite and Kuhn, 2020*), are the effects of locomotion on dLGN responses inherited via feedback from cortex? To test this hypothesis, we next focused on only those movie trials in which feedback was suppressed by V1 PV+ photostimulation and repeated the separation according to locomotion (*Figure 5g–h*). These analyses revealed that effects of locomotion on the responses to our movies persisted, even if CT feedback was suppressed (*Figure 5i–l*; firing rate: 9.7 vs 7.6 spikes/s; LMM: $F_{1,64.8} = 71.1$, $p = 5.2 \times 10^{-12}$; burst ratio: 0.081 vs 0.11 spikes/s; LMM: $F_{1,68.1} = 19.5$, $p = 3.7 \times 10^{-5}$; sparseness: 0.47 vs 0.56; LMM: $F_{1,179.5} = 54.7$, $p = 5.1 \times 10^{-12}$; reliability: 0.14 vs 0.18; LMM: $F_{1,175.7} = 24.9$, $p = 1.5 \times 10^{-6}$).

Besides running, another often-used indicator for behavioral state is pupil size (*Reimer et al., 2014*; *Vinck et al., 2015*; *Erisken et al., 2014*). Indexing arousal via pupil size, however, is challenging for movie stimuli, whose fluctuations in luminance will themselves drive changes in pupil size (*Figure 5—figure supplement 2a*). To test whether locomotion-independent, pupil-indexed arousal also modulates dLGN responses and whether this modulation depends on CT feedback, we exploited methods initially proposed by *Reimer et al., 2014*, focusing on periods within the movie when the animal was sitting and assuming that the average change in pupil size over multiple movie repetitions was due to luminance changes in the movie, while the variability around this average reflected trial-by-trial differences in behavioral state (*Figure 5—figure supplement 2b-g*). Recapitulating our running-related results, we found that both with CT feedback intact and during V1 suppression, response periods with faster than average pupil dilation (or slower than usual constriction; top quartile pupil change) were associated with higher firing rates, while periods with faster than usual pupil constriction (or slower than usual dilation; bottom quartile pupil change) were associated with lower firing rates (*Figure 5—figure supplement 2b-c*). In contrast, response reliability and SNR were not significantly different during periods of rapid dilation vs. rapid constriction, regardless of photostimulation condition (*Figure 5—figure supplement 2d-g*).

Finally, to further test the relationship between effects of behavioral state and CT feedback, we directly compared CT feedback and running-related modulations on a neuron-by-neuron basis. We focused on experiments with naturalistic movies, because this was the condition in which we observed robust effects of both CT feedback and behavioral state (for a related analysis with gratings and qualitatively similar results, see *Figure 6—figure supplement 1a*). First, we hypothesized that if effects of locomotion on dLGN responses were inherited from primary visual cortex, such effects should vanish during V1 suppression (*Figure 6*,a$_0$). However, consistent with the observations shown in *Figure 5i–l*, even during V1 suppression, running-related modulations were significantly different from 0 (firing rate run modulation index (RMI): $0.18 \pm 0.06$; burst ratio: $-0.17 \pm 0.1$; sparseness: $-0.12 \pm 0.04$; reliability: $-0.11 \pm 0.09$; *Figure 6*,a$_{1-4}$). In fact, the degree of running modulation was correlated between control conditions with feedback intact and V1 suppressed (firing rate: slope of $0.51 \pm 0.12$; burst ratio: slope of $0.38 \pm 0.2$; sparseness: slope of $0.44 \pm 0.14$; reliability: slope of $0.50 \pm 0.15$; *Figure 6*,a$_{1-4}$). Interestingly, for firing rates and burst ratios, locomotion effects were slightly stronger, on average, with CT feedback intact compared to V1 suppression (firing rate RMI: 0.23 vs 0.20; LMM: $F_{1,168.3} = 4.3$, $p = 0.04$, *Figure 6*, a$_1$; burst ratio RMI: $-0.25$ vs. $-0.17$; LMM: $F_{1,154.7} = 6.3$, $p = 0.013$, *Figure 6*, a$_2$), indicating that these two modulatory influences likely interact.

We next tested the hypothesis that CT feedback might have a stronger impact during active behavioral states than during quiescence. Indeed, it has previously been shown that during brain states associated with anesthesia, the responsiveness of feedback circuits is particularly reduced (*Briggs and Usrey, 2011*; *Makino and Komiyama, 2015*; *Keller et al., 2020*). One might therefore predict that during quiescence, if feedback circuits were already completely disengaged, we should not be able to observe further effects of V1 suppression (*Figure 6*, b$_0$). This was clearly not the case, because CT feedback effects were correlated across behavioral states (firing rate: slope of $0.72 \pm 0.10$; burst ratio: slope of $0.34 \pm 0.15$; sparseness: slope of $0.85 \pm 0.12$; reliability: slope of $0.43 \pm 0.14$; *Figure 6*, b$_{1-4}$). In addition, and similar to the slightly stronger run modulation with feedback left intact, we discovered

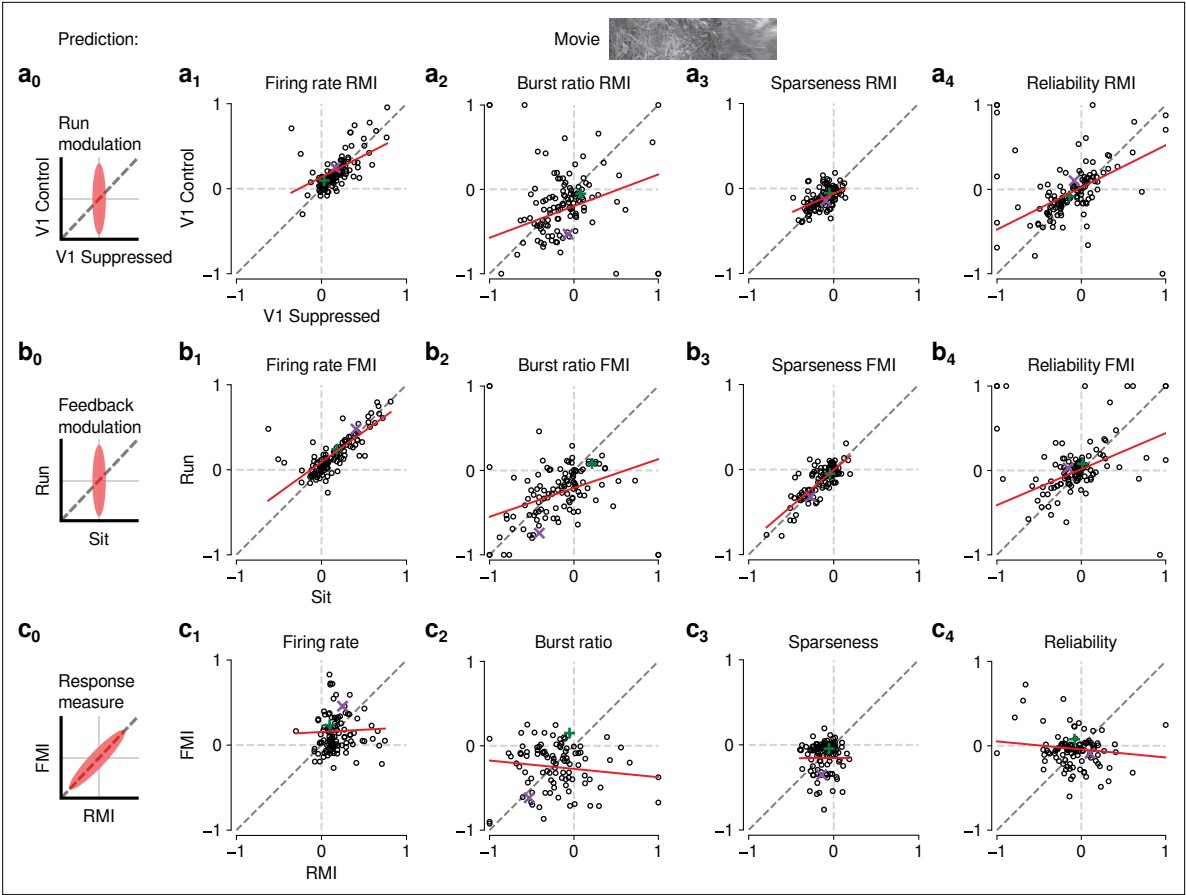

**Figure 6.** The effects of CT feedback and locomotion on movie responses are largely independent. ($a_0$–$c_0$) Predicted relationships between modulation indices and response measures in different conditions, assuming dependence in the effects of CT feedback and locomotion. (a) Comparison of modulation by running (RMI) during CT feedback intact and V1 suppression for firing rates ($a_1$), burst ratio ($a_2$), sparseness ($a_3$), and reliability ($a_4$). Running effects were quantified with a run modulation index (RMI), where RMI = (running − sitting)/(running + sitting). (b) Comparison of modulation by CT feedback (FMI) during locomotion and stationary periods for firing rates ($b_1$), burst ratio ($b_2$), sparseness ($b_3$), and reliability ($b_4$). (c) Comparison of modulation by feedback (FMI) and modulation by running (RMI) for firing rates ($c_1$), burst ratio ($c_2$), sparseness ($c_3$), and reliability ($c_4$). *Red*: LMM fit. *Green, purple*: example neurons from **Figure 2a and b**.

The online version of this article includes the following figure supplement(s) for figure 6:

**Figure supplement 1.** The effects of CT feedback and locomotion on responses to gratings are also largely independent.

a locomotion-dependent CT feedback effect for firing rates and burst ratios: CT feedback effects were slightly stronger, on average, during locomotion than during quiescence (firing rate FMI: 0.18 vs 0.15; LMM: $F_{1,172.8} = 3.5$, $p = 0.065$; **Figure 6**, $b_1$; burst ratio FMI: −0.27 vs. −0.19; LMM: $F_{1,166.9} = 6.8$, $p = 0.0097$; **Figure 6**, $b_2$). This subtle interaction between behavioral state and CT feedback effects might relate to a previous finding, where careful dissection of brain states by depth of anesthesia had already suggested that the effects of transient cortical inactivation on dLGN responses were more evident during lighter anesthesia, that is, during desynchronized cortical activity (**Wörgötter et al., 2002**). However, our ability to observe effects of V1 suppression in dLGN while the animal was stationary suggests that CT feedback circuits are engaged even under conditions of behavioral quiescence.

Finally, if modulations by CT feedback and behavioral state exploited the same circuitry, neurons experiencing strong modulation by V1 suppression should also be strongly affected by locomotion (**Figure 6**, $c_0$). Contrary to this prediction, we found that effects of CT feedback (FMI) and behavioral state (RMI) were uncorrelated (firing rate: slope of $0.054 \pm 0.13$; burst ratio: slope of $−0.1 \pm 0.13$; sparseness: slope of $0.005 \pm 0.23$; reliability: slope of $−0.095 \pm 0.12$; **Figure 6**$c_{1-4}$). Together, these comparisons demonstrate that effects of behavioral state associated with locomotion and effects of CT feedback are largely independent.

## Discussion

In this study, we used naturalistic movies to reveal that corticothalamic feedback and behavioral state can have robust effects on dLGN responses. We found that V1 suppression during movie presentation reduces the gain of time-varying dLGN firing rates, and leads to increases in bursting, sparseness and trial-to-trial reliability. The effects of CT feedback seem to be stimulus-specific, as V1 suppression led to more consistent and therefore stronger overall effects on firing rates for naturalistic movies than for gratings. Interestingly, the signatures of CT feedback closely resembled those of behavioral state. However, we found their effects during movie viewing to be largely independent, demonstrating that behavioral modulations of dLGN activity are not simply inherited from cortex. Overall, our findings highlight that dLGN responses to naturalistic movies can be reliably modulated by two extra-retinal sources – cortical feedback and behavioral state – which likely exert their influences via largely separate neural circuits.

### Manipulation of CT feedback

To manipulate CT feedback, we chose a potent, yet global, V1 suppression approach based on optogenetic activation of ChR2 expressed in local PV+ inhibitory interneurons (*Lien and Scanziani, 2013*; *Li et al., 2013*; *King et al., 2016*; *Olsen et al., 2012*; *Wiegert et al., 2017*). While silencing by excitation of inhibitory interneurons can exploit the robust effects of GABA-mediated inhibition in cortical circuits, it comes with a limitation in specificity. Hence, in addition to the direct L6 → thalamus circuit, indirect polysynaptic effects might be exerted via alternative routes. One example is L5 corticofugal pyramidal cells projecting to the superior colliculus (SC), where tectogeniculate neurons in the superficial layers provide retinotopically organized, driving inputs to the dorsolateral shell region of the dLGN (*Bickford et al., 2015*). To address this lack of specificity, in control experiments, we replaced photoactivation of PV +neurons with direct, selective suppression of V1 Ntsr1 +neurons, encompassing the population of L6 CT pyramidal cells (*Kim et al., 2014*; *Bortone et al., 2014*). Since photosuppression via the light-gated chloride channel stGtACR2 (*Mahn et al., 2018*) did not alter any of our conclusions regarding the effects of CT feedback on dLGN responses, we assume that the effects of V1 suppression to a large degree reflect the specific impact of the L6 CT circuit. L6 CT neurons, however, have an intracortical axon collateral making privileged connections with a translaminar PV +interneuron subtype in L6 (*Frandolig et al., 2019*; *Bortone et al., 2014*), which in turn strongly regulates the gain of the entire V1 column (*Olsen et al., 2012*; *Bortone et al., 2014*; *Frandolig et al., 2019*), so that even with such specific suppression, polysynaptic effects cannot be excluded. However, since suppression of L6 CT neurons increases the gain in V1 (*Olsen et al., 2012*), and since this is the opposite of the global effects of V1 suppression via PV +activation, L6 CT gain modulation of V1 seems unlikely to drive our effects. Nevertheless, decisively ruling out alternative circuits would require the selective suppression of L6 CT axon terminals at the thalamic target.

Cortical layer 6 is well known for its particularly high diversity of neuronal cell types (*Briggs, 2010*). Even within the population of L6 CT pyramidal cells there is heterogeneity, with at least two subtypes defined by morphology (*Frandolig et al., 2019*; *Tasic et al., 2016*; *Gouwens et al., 2019*; *Augustinaite and Kuhn, 2020*), three subtypes defined by electrophysiology and morphology (*Gouwens et al., 2019*), and four major subtypes defined by transcriptomics (*Tasic et al., 2016*; *Gouwens et al., 2019*). Whether these subtypes mediate different aspects of feedback modulations is currently unknown. In the visual system of primates and carnivores, CT feedback circuits seem to be organized into distinct streams (*Briggs et al., 2016*; *Hasse et al., 2019*; *Briggs and Usrey, 2009*) whose functional organization mimics that of the feedforward streams. Whether the known subtypes in mice can convey independent, stream-specific information is currently unknown, partly because already at the level of feedforward processing, the notion of streams in mouse dLGN is a matter of ongoing debate (*Chen et al., 2016*; *Denman and Contreras, 2016*; *Morgan et al., 2016*; *Chen et al., 2016*; *Zhuang et al., 2019*), and dLGN response properties are diverse (*Piscopo et al., 2013*; *Román Rosón et al., 2019*; *Liang et al., 2018*). Our own assessment of CT feedback effects revealed few systematic differences for various dLGN cell-type classifications. Such an absence of differences, however, is not surprising, because our optogenetic circuit manipulations non-specifically suppressed all L6 CT neuron subtypes. Once genetic targeting of L6 CT subtypes will become possible, it will be important to test the stream-specificity of CT feedback in the mouse.

## CT feedback effects on gain, reliability, and bursting

Our analyses of the time-varying firing rates in response to naturalistic movies revealed that V1 suppression results in a robust decrease of geniculate response gain. Divisive effects of CT feedback suppression have also been previously reported for contrast response functions of parvocellular dLGN neurons in anesthetized macaques (*Przybyszewski et al., 2000*). A crucial element to produce gain modulations seems to be changes in the level of synaptically driven $V_m$ fluctuations, often called 'synaptic noise' (*Hô and Destexhe, 2000*; *Shu et al., 2003*; *Chance et al., 2002*). Indeed, in vivo V1 recordings suggest that the combined impact of changes in $V_m$ fluctuations, input resistance, and depolarization is needed to produce gain changes (*Cardin et al., 2008*). These cellular properties are altered by both feedback (*Chance et al., 2002*) and neuromodulation (*Disney et al., 2007*), not only in cortex (*Ferguson and Cardin, 2020*) but also in the corticothalamic system (*Béhuret et al., 2015*; *Augustinaite et al., 2014*). Here, 'synaptic noise' together with varying degrees of T-type channel recruitment has been shown to change the slope of the input-output function and alter the temporal filtering characteristics of thalamic relay cells (*Wolfart et al., 2005*; *Béhuret et al., 2015*). Thus, by providing variable synaptic input and affecting membrane depolarization, for example, through NMDA plateau potentials (*Augustinaite et al., 2014*), CT feedback might be in a prime position to dynamically tune the gain of the thalamic relay.

In addition to potentially contributing to the observed gain modulations, 'synaptic noise' from CT feedback may also help explain the less precise and less reliable dLGN responses we observed when feedback was left intact. Specifically, V1 neurons are known to exhibit about double the trial-to-trial variability of simultaneously recorded dLGN neurons (*Kara et al., 2000*), and eliminating variable cortical input might unmask the even greater reliability of feed-forward retinal inputs (*Kara et al., 2000*).

Our analyses of movie and grating response characteristics showed that V1 suppression robustly and consistently biased geniculate activity toward burst firing mode. Burst firing mode occurs when dLGN neurons undergo sustained ($\geq 100$ ms) hyperpolarization (*Sherman, 2001*), which allows for the de-inactivation of low-threshold T-type calcium channels abundant in thalamus (*Jahnsen and Llinás, 1984*). Such 'calcium bursts' can only be unequivocally separated from high-frequency firing in intracellular recordings or calcium imaging, but can be inferred in extracellular recordings, such as ours, by imposing a minimum duration of 100 ms of silence preceding a high-frequency ( < 4 ms ISI) firing event (*Lu et al., 1992*). Previous in vivo intracellular recordings in cat dLGN have revealed that cortical ablation can hyperpolarize the resting membrane potential of dLGN relay cells by ~9 mV, enough to push them into burst-firing mode (*Dossi et al., 1992*). Conversely, direct optogenetic activation of L6 CT neurons in primary somatosensory cortex has been shown to decrease burst mode firing (*Mease et al., 2014*), potentially mediated by NMDA plateau potentials as observed in slice recordings (*Augustinaite et al., 2014*). In burst firing mode, reminiscent of the effects we observed during V1 suppression, dLGN spontaneous activity is low (*Sherman, 2001*), stimulus-evoked responses show phase-advance (*Lu et al., 1992*; *Alitto et al., 2005*) and high trial-to-trial reliability (*Alitto et al., 2005*). The increase in trial-to-trial response reliability we observed during V1 suppression might therefore be explained not only by the removal of a more variable input as mentioned above (*Kara et al., 2000*), but also by a shift towards burst mode, where retinogeniculate communication efficacy is elevated (*Alitto et al., 2019*).

Theories about the function of thalamic firing modes can provide a useful framework for interpreting the effects of CT feedback we observed here, in particular since the greater precision and trial-to-trial reliability of responses during V1 suppression might be unexpected at first glance. Thalamic burst mode is often linked with 'inattentive states', where the sudden appearance or change of a visual stimulus from non-preferred to preferred RF contents (*Lesica and Stanley, 2004*; *Lesica et al., 2006*; *Wang et al., 2007*) can reliably trigger a thalamic burst. Bursting is associated with high signal-to-noise, well-suited for stimulus detection (*Sherman, 2001*; *Whitmire et al., 2016*). In addition, thalamic burst mode is known to augment the efficacy of retinal input to drive spiking in dLGN (*Alitto et al., 2019*), and increases the probability of relay between thalamus and cortex (*Swadlow and Gusev, 2001*). This in turn might lead to depolarizing CT feedback, switching the thalamus to tonic mode and allowing more faithful, linear relay of information with a higher dynamic range, better suited for encoding of more finely graded details (*Sherman, 2001*; *Béhuret et al., 2015*). Such a 'wake-up-call' for cortex (*Sherman, 2001*; *Lesica and Stanley, 2004*) could represent a neural implementation

of bottom-up attention in dLGN (*Hochstein and Ahissar, 2002*). To understand if CT feedback is indeed recruited for detailed perceptual analyses, an essential next step would be to measure the activity of L6 CT neurons under behaviorally relevant conditions. Interestingly, in the auditory system, activation of L6 CT feedback has been shown to influence sound perception, with enhancements of sound detection or discrimination behavior, depending on the relative timing between CT spiking and stimulus onset (*Guo et al., 2017*). Beyond having a broad impact on coding regimes and transmission, bursting in thalamus is also known to have specific computational properties, such as efficiently encoding high- and low-frequency information in parallel (*Mease et al., 2017*).

## Stimulus-dependence of CT feedback effects

So far, most studies using naturalistic stimuli to probe dLGN responses have been performed in anesthetized animals and have not considered CT feedback (*Dan et al., 1996*; *Lesica and Stanley, 2004*; *Lesica et al., 2006*; *Lesica et al., 2007*; *Wang et al., 2007*; *Mante et al., 2005*). Similarly, most studies investigating the impact of CT feedback have relied on artificial stimuli (*Olsen et al., 2012*; *Denman and Contreras, 2015*; *Wang et al., 2006*; *Murphy and Sillito, 1987*). Comparing the effects of CT feedback during naturalistic movies and gratings, we found evidence that CT feedback modulates firing rates at the geniculate level in a stimulus-dependent fashion. What could be the relevant difference? For artificial stimuli, such as gratings and bars, it has long been known that CT feedback can enhance dLGN surround suppression by increasing responses to small stimuli and reducing responses to large stimuli (*Born et al., 2021*; *McClurkin and Marrocco, 1984*; *Murphy and Sillito, 1987*; *Jones et al., 2012*; *Wang et al., 2018*; *Cudeiro and Sillito, 1996*; *Andolina et al., 2013*; *Hasse and Briggs, 2017*; *Webb et al., 2002*). Such CT feedback-mediated enhancement of surround suppression might result from recruitment of a more narrow direct excitatory and a wider indirect inhibitory CT feedback component according to grating size (*Born et al., 2021*), with the balance shifting more towards direct excitation for small gratings and more towards indirect inhibition for large gratings. Size, however, is likely not the only determinant of relative recruitment of CT feedback circuits: for instance, V1 ablation or pharmacological suppression in anesthetized cats leads to more prominent reductions of dLGN surround suppression for iso- vs. cross-oriented gratings (*Cudeiro and Sillito, 1996*; *Sillito et al., 1993*), suggesting an additional role of stimulus context. For naturalistic stimuli with complex context, measurements in area V1 have already demonstrated that surround suppression is generally lower than for iso-oriented gratings, and is flexibly invoked depending on the specific statistics in the RF center and surround (*Coen-Cagli et al., 2015*). The differential effect of CT feedback on dLGN firing rates for full-screen naturalistic movies and iso-oriented gratings observed in our study might therefore be parsimoniously explained by differences in the relative strength of direct excitatory and indirect inhibitory CT feedback. It would be of prime interest to measure, in future experiments, size tuning curves with and without CT feedback using different stimuli, such as naturalistic movies, iso- and cross-oriented gratings. Given our results, we predict that CT feedback would affect firing rate responses to full-screen cross-oriented gratings more similarly to full-screen naturalistic movies than would iso-oriented gratings. Alternatively, CT feedback might change firing rates more consistently for lower contrast stimuli, such as our movies, where additional top-down inputs might be helpful for detection or discrimination.

## Relationship between CT feedback and behavioral state

By measuring the effects of V1 suppression on movie responses during different behavioral states, and by measuring effects of behavioral state with and without CT feedback, we found that behavioral state and CT feedback had similar effects on dLGN responses, but seemed to operate via largely separate circuits. The lack of substantial dependence between effects of CT feedback and behavioral state on responses to our naturalistic movies is remarkable: neuromodulation accompanying changes in behavioral state will affect cortical layer 6, which receives dense cholinergic afferents from basal forebrain (*Radnikow and Feldmeyer, 2018*). Accordingly, in slice recordings, upon bath application of ACh, mouse V1 L6 CT neurons increase action potential firing (*Sundberg et al., 2018*). Potentially related, many V1 L6 CT neurons themselves increase activity during locomotion or arousal (*Augustinaite and Kuhn, 2020*; *Swadlow and Weyand, 1987*). Together, these studies would predict that effects of behavioral state should be augmented during CT feedback. Indeed, two recent studies investigating the interactions between CT feedback and arousal reported, during suppression of CT feedback,

less correlation between dLGN firing and pupil size (*Molnár et al., 2021*), and a loss of effects of behavioral state on dLGN tuning curves for temporal and spatial frequency, but not for spontaneous activity (*Reinhold et al., 2021*). Together with other findings more consistent with our results (*Murata and Colonnese, 2018*; *Nestvogel and McCormick, 2022*; *Schröder et al., 2020*), this discrepancy suggests that the degree to which effects of behavioral state in dLGN might be dependent on cortex is not fully understood.

If not inherited from CT feedback, which alternative circuits could mediate the effects of behavioral state in dLGN (*Erisken et al., 2014*; *Aydın et al., 2018*; *Williamson et al., 2015*)? Locomotion is accompanied by arousal (*Vinck et al., 2015*), which in turn involves various neuromodulatory influences [reviewed in *Zagha and McCormick, 2014*]. For instance, norepinephrine from the locus coeruleus (LC) and acetylcholine (ACh) from the midbrain are known to act directly on the thalamus [reviewed in *McCormick, 1992*; *Lee and Dan, 2012*] and could drive some of the arousal-related depolarizing effects on firing rate independent of cortical feedback, for instance by blocking a long-lasting $Ca^{2+}$-dependent $K^+$ current (*Sherman and Koch, 1986*). In addition, electrical stimulation of the LC (*Holdefer and Jacobs, 1994*) and the parabrachial region (PBR) (*Lu et al., 1993*) within the mesencephalic locomotor region (MLR), and direct application of noradrenergic (*Funke et al., 1993*) and cholinergic (*McCormick, 1992*; *Sillito et al., 1983*) agonists within dLGN, are sufficient to reduce thalamic burst mode firing. Finally, at least part of the locomotion effects in dLGN might also be related to modulations of retinal output (*Schröder et al., 2020*; *Liang et al., 2020*). Indeed, two-photon calcium imaging of retinal ganglion cell boutons in dLGN (*Liang et al., 2020*) and SC (*Schröder et al., 2020*) revealed that their activity can be modulated by locomotion, albeit with an overall suppressive effect. In future studies, it will be key to further dissect the contributions of retinal, cortical and potentially collicular modulations, and the different neuromodulatory sources of behavioral state-related modulations in thalamic targets.

## Materials and methods

### Key resources table

| Reagent type (species) or resource | Designation | Source or reference | Identifiers | Additional information |
|---|---|---|---|---|
| Recombinant DNA reagent | pAAV EF1a.DIO.hChR2(H134R)- eYFP.WPRE.hGH | Addgene | #20298-AAV9 | |
| Recombinant DNA reagent | pAAV hSyn1-SIO-stGtACR2- FusionRed | Addgene | #105,677 | |
| Strain, strain background (*Mus musculus*) | B6;129P2-*Pvalb*[tm1(cre)Arbr]/J | Jackson Laboratory | #008069 | PV-Cre, Pvalb-Cre |
| Strain, strain background (*Mus musculus*) | B6.FVB(Cg)-Tg(*Ntsr1*-cre) GN220Gsat/Mmcd | MMRRC | #030648-UCD | Ntsr1-Cre |
| Chemical compound, drug | Metamizole | MSD Animal Health | Vetalgin | 200 mg/kg |
| Chemical compound, drug | Buprenorphine | Bayer | Buprenovet | 0.1 mg/kg |
| Chemical compound, drug | Lidocaine hydrochloride | bela-pharm | | 2 % |
| Chemical compound, drug | Meloxicam | Böhringer Ingelheim | Metacam | 2 mg/kg |
| Chemical compound, drug | Isoflurane | CP Pharma | | in oxygen |
| Chemical compound, drug | Bepanthen | Bayer | | eye ointment |
| Chemical compound, drug | DAPI-containing mounting medium | Vector Laboratories Ltd | | |
| Chemical compound, drug | Vectashield DAPI H-1000 | Vector Laboratories Ltd | | |
| Chemical compound, drug | DiI | Invitrogen | | electrode stain |
| Software, algorithm | Python 3.6 | http://python.org | RRID:SCR_008394 | |
| Software, algorithm | R | *R Core Team, 2017* | RRID:SCR_001905 | |
| Software, algorithm | MATLAB R2019b | Mathworks | RRID:SCR_001622 | |
| Software, algorithm | EXPO | https://sites.google.com/a/nyu.edu/expo/home | | visual stimulus display |

*Continued on next page*

Continued

| Reagent type (species) or resource | Designation | Source or reference | Identifiers | Additional information |
|---|---|---|---|---|
| Software, algorithm | Kilosort | *Pachitariu et al., 2016* | RRID:SCR_016422 | |
| Software, algorithm | Spyke | *Spacek et al., 2009* | | |
| Software, algorithm | Fiji/ImageJ | NIH | RRID:SCR_003070 | |
| Software, algorithm | DataJoint | *Yatsenko et al., 2018* | RRID:SCR_014543 | |

## Surgical procedures

Experiments were carried out in 6 adult PV-Cre mice (median age at first recording session: 23.5 weeks; B6;129P2-*Pvalb*^tm1(cre)Arbr^/J; #008069, Jackson Laboratory) and 3 adult Ntsr1-Cre mice (median age: 29.4 weeks; B6.FVB(Cg)-Tg(*Ntsr1*-cre)GN220Gsat/Mmcd; #030648-UCD, MMRRC) of either sex. Thirty minutes prior to the surgical procedure, mice were injected with an analgesic (Metamizole, 200 mg/kg, sc, MSD Animal Health, Brussels, Belgium). To induce anesthesia, animals were placed in an induction chamber and exposed to isoflurane (5% in oxygen, CP-Pharma, Burgdorf, Germany). After induction of anesthesia, mice were fixated in a stereotaxic frame (Drill & Microinjection Robot, Neurostar, Tuebingen, Germany) and the isoflurane level was lowered (0.5–2% in oxygen), such that a stable level of anesthesia could be achieved as judged by the absence of a pedal reflex. Throughout the procedure, the eyes were covered with an eye ointment (Bepanthen, Bayer, Leverkusen, Germany) and a closed loop temperature control system (ATC 1000, WPI Germany, Berlin, Germany) ensured that the animal's body temperature was maintained at 37 ° C. At the beginning of the surgical procedure, an additional analgesic was administered (Buprenorphine, 0.1 mg/kg, sc, Bayer, Leverkusen, Germany) and the animal's head was shaved and thoroughly disinfected using iodine solution (Braun, Melsungen, Germany). Before performing a scalp incision along the midline, a local analgesic was delivered (Lidocaine hydrochloride, sc, bela-pharm, Vechta, Germany). The skin covering the skull was partially removed and cleaned from tissue residues with a drop of $H_2O_2$ (3%, AppliChem, Darmstadt, Germany). Using four reference points (bregma, lambda, and two points 2 mm to the left and to the right of the midline respectively), the animal's head was positioned into a skull-flat configuration. The exposed skull was covered with OptiBond FL primer and adhesive (Kerr dental, Rastatt, Germany) omitting three locations: V1 (AP: -2.8 mm, ML: -2.5 mm), dLGN (AP: -2.3 mm, ML: -2 mm), and a position roughly 1.5 mm anterior and 1 mm to the right of bregma, designated for a miniature reference screw (00–96 X 1/16 stainless steel screws, Bilaney) soldered to a custom-made connector pin. Two µL of the adeno-associated viral vector rAAV9/1. EF1a.DIO.hChR2(H134R)-eYFP.WPRE.hGH (Addgene, #20298-AAV9) was dyed with 0.3 µL fast green (Sigma-Aldrich, St. Louis, USA). After performing a small craniotomy over V1, in PV-Cre mice a total of ~ 0.5 µL of this mixture was injected across the entire depth of cortex (0.05 µL injected every 100 µm, starting at 1000 µm and ending at 100 µm below the brain surface), using a glass pipette mounted on a Hamilton syringe (SYR 10 µL 1701 RN no NDL, Hamilton, Bonaduz, Switzerland). In V1 of Ntsr1-Cre mice, we injected 0.35 µL of stGtACR2 (pAAV_hSyn1-SIO-stGtACR2-FusionRed, Addgene, #105677; 0.05 µL injected every 100 µm, starting at 1000 µm and ending at 500 µm below the brain surface). A custom-made lightweight stainless steel head bar was positioned over the posterior part of the skull such that the round opening in the bar was centered on V1/dLGN. The head bar was attached with dental cement (Ivoclar Vivadent, Ellwangen, Germany) to the primer/adhesive. The opening was later filled with the silicone elastomer sealant Kwik-Cast (WPI Germany, Berlin, Germany). At the end of the procedure, an antibiotic ointment (Imex, Merz Pharmaceuticals, Frankfurt, Germany) or iodine-based ointment (Braunodivon, 10%, B. Braun, Melsungen, Germany) was applied to the edges of the wound and a long-term analgesic (Meloxicam, 2 mg/kg, sc, Böhringer Ingelheim, Ingelheim, Germany) was administered and for 3 consecutive days. For at least 5 days post-surgery, the animal's health status was assessed via a score sheet. After at least 1 week of recovery, animals were gradually habituated to the experimental setup by first handling them and then simulating the experimental procedure. To allow for virus expression, neural recordings started no sooner than 3 weeks after injection. On the day prior to the first day of recording, mice were fully anesthetized using the same procedures as described for the initial surgery, and

a craniotomy (ca. 1.5 mm$^2$) was performed over dLGN and V1 and re-sealed with Kwik-Cast (WPI Germany, Berlin, Germany). As long as the animals did not show signs of discomfort, the long-term analgesic Metacam was administered only once at the end of surgery, to avoid any confounding effect on experimental results. Recordings were performed daily and continued for as long as the quality of the electrophysiological signals remained high.

## Electrophysiological recordings, optogenetic suppression of V1, perfusion

Head-fixed mice were placed on an air-cushioned Styrofoam ball, which allowed the animal to freely move. Two optical computer mice interfaced with a microcontroller (Arduino Duemilanove) sampled ball movements at 90 Hz. To record eye position and pupil size, the animal's eye was illuminated with infrared light and monitored using a zoom lens (Navitar Zoom 6000) coupled with a camera (Guppy AVT camera; frame rate 50 Hz, Allied Vision, Exton, USA). Extracellular signals were recorded at 30 kHz (Blackrock microsystems). For each recording session, the silicon plug sealing the craniotomy was removed. For V1 recordings, a 32- or 64 channel silicon probe (Neuronexus, A1 × 32-5 mm-25-177, A1 × 32Edge-5mm-20–177 A32 or A1 × 64-Poly2-6mm-23s-160) was lowered into the brain to a median depth of 1025 µm. For dLGN recordings, a 32-channel linear silicon probe (Neuronexus A1 × 32Edge-5mm-20–177 A32) was lowered to a depth of ~2300–3611 µm below the brain surface. We judged recording sites to be located in dLGN based on the characteristic progression of RFs from upper to lower visual field along the electrode shank (*Piscopo et al., 2013*, *Figure 1—figure supplement 1b*), the presence of responses strongly modulated at the temporal frequency of the drifting gratings (F1 response), and the preference of responses to high temporal frequencies (*Grubb and Thompson, 2003*; *Piscopo et al., 2013*). For *post hoc* histological reconstruction of the recording site, the electrode was stained with DiI (Invitrogen, Carlsbad, USA) for one of the final recording sessions.

For photostimulation of V1 PV +inhibitory interneurons or photosuppression of V1 L6CT neurons, an optic fiber (910 µm diameter, Thorlabs, Newton, USA) was coupled to a light-emitting diode (LED, center wavelength 470 nm, M470F1, Thorlabs, Newton, USA; or center wavelength 465 nm, LEDC2_465/635_SMA, Doric Lenses, Quebec, Canada) and positioned with a micromanipulator less than 1 mm above the exposed surface of V1. A black metal foil surrounding the tip of the head bar holder prevented most of the photostimulation light from reaching the animal's eyes. To ensure that the photostimulation was effective, the first recording session for each mouse was carried out in V1. Only if the exposure to light reliably induced suppression of V1 activity was the animal used for subsequent dLGN recordings. For gratings, photostimulation started either 0.1 s before stimulus onset and ended 0.1 s after stimulus offset (2 experiments), or photostimulation started 0.3 s before stimulus onset and ended 0.2 s after stimulus offset (11 experiments), or photostimulation started 0.3 s before stimulus onset and ended 0.45 s after stimulus offset (12 experiments). For movie clips, photostimulation started either 0.1 s before stimulus onset and ended 0.1 s after stimulus offset (2 experiments), or photostimulation started 0.3 s before stimulus onset and ended 0.45 s after stimulus offset (45 experiments). LED light intensity was adjusted on a daily basis to evoke reliable effects (median intensity: 13.66 mW/mm$^2$ for activating ChR2 in PV-Cre mice, and 10.84 mW/mm$^2$ for activating stGtACR2 in Ntsr1-Cre mice, as measured at the tip of the optic fiber). Since the tip of the fiber never directly touched the surface of the brain, and since the clarity of the surface of the brain varied (generally decreasing every day following the craniotomy), the light intensity delivered even to superficial layers of V1 was inevitably lower. Importantly, changes in dLGN firing rates induced by V1 suppression (FMI, see below) did not differ, on average, from those induced by behavioral state (RMI, see below) (firing rate: FMI 0.20 vs. RMI 0.15, LMM: $F_{1,145.7} = 3.02$, $p = 0.08$; burst ratio: FMI $-0.27$ vs. RMI $-0.28$, $F_{1,124.0} = 0.002$, $p = 0.97$; sparseness: FMI $-0.12$ vs. RMI $-0.14$, $F_{1,144.9} = 1.03$, $p = 0.31$; reliability: FMI $-0.084$ vs. $-0.037$, $F_{1,183.0} = 1.96$, $p = 0.16$; *Figure 6c*), indicating that optogenetic stimulation effects were not outside the physiological range.

After the final recording session, mice were first administered an analgesic (Metamizole, 200 mg/kg, sc, MSD Animal Health, Brussels, Belgium) and following a 30 min latency period were transcardially perfused under deep anesthesia using a cocktail of Medetomidin (Domitor, 0.5 mg/kg, Vetoquinol, Ismaning, Germany), Midazolam (Climasol, 5 mg/kg, Ratiopharm, Ulm, Germany) and Fentanyl (Fentadon, 0.05 mg/kg, Dechra Veterinary Products Deutschland, Aulendorf, Germany) (ip). A few animals, which were treated according to a different license, were anesthetized with sodium

pentobarbital (Narcoren, 400 mg/kg, ip, Böhringer Ingelheim, Ingelheim, Germany). Perfusion was first done with Ringer's lactate solution followed by 4% paraformaldehyde (PFA) in 0.2 M sodium phosphate buffer (PBS).

## Histology

To verify recording site and virus expression, we performed histological analyses. Brains were removed, postfixed in PFA for 24 hr, and then rinsed with and stored in PBS at 4 °C. Slices (40 μm) were cut using a vibratome (Leica VT1200 S, Leica, Wetzlar, Germany), stained with DAPI solution before (DAPI, Thermo Fisher Scientific; Vectashield H-1000, Vector Laboratories) or after mounting on glass slides (Vectashield DAPI), and coverslipped. A fluorescent microscope (BX61, Olympus, Tokyo, Japan) was used to inspect slices for the presence of yellow fluorescent protein (eYFP) and DiI. Recorded images were processed using FIJI (*Rueden et al., 2017*; *Schindelin et al., 2012*).

## Visual stimulation

Visual stimuli were presented on a liquid crystal display (LCD) monitor (Samsung SyncMaster 2233RZ, 47 × 29 cm, 1680 × 1050 resolution at 60 Hz, mean luminance 50 cd/m$^2$) positioned at a distance of 25 cm from the animal's right eye (spanning ~ 108 × 66°, small angle approximation) using custom written software (EXPO, https://sites.google.com/a/nyu.edu/expo/home). The display was gamma-corrected for the presentation of artificial stimuli, but not for movies (see below).

To measure receptive fields (RFs), we mapped the ON and OFF subfields with a sparse noise stimulus. The stimulus consisted of nonoverlapping white and black squares on a square grid, each flashed for 200ms. For dLGN recordings, the square grid spanned 60° on a side, while individual squares spanned 5° on a side. For a single experiment, the vertical extent was reduced to 50°. For subsequent choices of stimuli, RF positions and other tuning preferences were determined online after each experiment based on multiunit activity, that is high-pass filtered signals crossing a threshold of 4.5–6.5 SD.

We measured single unit orientation preference by presenting full-screen, full-contrast drifting sinusoidal gratings of either 12 (23 experiments) or 8 (2 experiments) different, pseudo-randomly interleaved orientations (30° or 45° steps). For dLGN recordings, spatial frequency was either 0.02 cyc/° (17 experiments) or 0.04 cyc/° (8 experiments) and temporal frequency was either 2 Hz (2 experiments) or 4 Hz (23 experiments). One blank condition (i.e. mean luminance gray screen) was included to allow measurements of spontaneous activity. The stimulus duration was either 2 s (23 experiments) or 5 s (2 experiments), with an interstimulus interval (ISI) of 2.4 s (21 experiments) or 1.25 s (2 experiments). For two Ntsr1-Cre experiments, ISIs varied and were either 0.58 s or 1.09 s.

For laminar localization of neurons recorded in V1, we presented a full-screen, contrast-reversing checkerboard at 100% contrast, with a spatial frequency of either 0.01 cyc/° (2 experiments) or 0.02 cyc/° (5 experiments) and a temporal frequency of 0.5 cyc/s.

Movies were acquired using a hand-held consumer-grade digital camera (Canon PowerShot SD200) at a resolution of 320 × 240 pixels and 60 frames/s. Movies were filmed close to the ground in a variety of wooded or grassy locations in Vancouver, BC, and contained little to no forward/backward optic flow, but did contain simulated gaze shifts (up to 275°/s), generated by manual camera movements (for example movies, see *Figure 1—video 1* and *Figure 1—video 2*). Focus was kept within 2 m and exposure settings were set to automatic. The horizontal angle subtended by the camera lens was 51.6°. No display gamma correction was used while presenting movies, since consumer-grade digital cameras are already gamma corrected for consumer displays (*Poynton, 1998*). For presentation, movies were cut into 5 s clips and converted from color to grayscale. Movie clips were presented full-screen with an ISI of 1.25 s (43 experiments). For two Ntsr1-Cre experiments, ISIs varied and were either 0.58 s or 1.08 s.

## Spike sorting

To obtain single unit activity from extracellular recordings, we used the open source, Matlab-based, automated spike sorting toolbox Kilosort (*Pachitariu et al., 2016*). Resulting clusters were manually refined using Spyke (*Spacek et al., 2009*), a Python application that allows the selection of channels and time ranges around clustered spikes for realignment, as well as representation in 3D space using dimension reduction (multichannel PCA, ICA, and/or spike time). In 3D, clusters were then further split

via a gradient-ascent based clustering algorithm (GAC) (*Swindale and Spacek, 2014*). Exhaustive pairwise comparisons of similar clusters allowed the merger of potentially over-clustered units. For subsequent analyses, we inspected autocorrelograms and mean voltage traces, and only considered units that displayed a clear refractory period and a distinct spike waveshape. All further analyses were carried out using the DataJoint framework (*Yatsenko et al., 2018*) with custom-written code in Python.

## Response characterization

We used current source density (CSD) analysis for recordings in area V1 to determine the laminar position of electrode contacts. To obtain the LFP data we first down-sampled the signal to 1 kHz before applying a bandpass filter (4–90 Hz, 2nd-order Butterworth filter). We computed the CSD from the second spatial derivative of the local field potentials (*Mitzdorf, 1985*), and assigned the base of layer 4 to the contact that was closest to the earliest CSD polarity inversion. The remaining contacts were assigned to supragranular, granular and infragranular layers, assuming a thickness of ~1 mm for mouse visual cortex (*Heumann et al., 1977*).

In recordings targeting dLGN, we used the envelope of multi-unit spiking activity (MUAe) (*van der Togt et al., 2005*) to determine RF progression (*Figure 1—figure supplement 1b*). Briefly, we full-wave rectified the high-pass filtered signals (cutoff frequency: 300 Hz, 4th-order non-causal Butterworth filter) before performing common average referencing by subtracting the median voltage across all channels in order to eliminate potential artifacts (e.g. movement artifacts). We then applied a low-pass filter (cutoff frequency: 500 Hz, Butterworth filter) and down-sampled the signal to 2 kHz. Recording sessions for which RFs did not show the retinotopic progression typical of dLGN (*Figure 1—figure supplement 1b*; *Piscopo et al., 2013*) were excluded from further analysis.

Each unit's peristimulus time histogram (PSTH, i.e. the response averaged over trials) was calculated by convolving a Gaussian of width $2\sigma$ = 20 ms with the spike train collapsed across all trials, separately for each condition.

We defined bursts according to *Lu et al., 1992*, which required a silent period of at least 100ms before the first spike in a burst, followed by a second spike with an interspike interval < 4 ms. Imposing the silent period was found to be crucial for separating dLGN 'low threshold calcium bursts' from high-frequency firing in extracellular recordings (*Lu et al., 1992*); note however, that 'low-threshold calcium bursts' can only be unequivocally detected in intracellular recordings or calcium imaging. Any subsequent spikes with preceding interspike intervals < 4ms were also considered to be part of the burst. All other spikes were regarded as tonic. We computed a burst ratio (the number of burst spikes divided by the total number of spikes) and compared this ratio in conditions with CT feedback intact vs. V1 suppression or during locomotion vs. stationary conditions. PSTHs for burst spikes were calculated by only considering spikes that were part of bursts before collapsing across trials and convolving with the Gaussian kernel (see above). PSTHs for non-burst spikes were calculated in an analogous way.

To quantify the effect of V1 suppression on various response properties, we defined the feedback modulation index (FMI) as

$$\text{FMI} = \frac{\text{feedback}-\text{suppression}}{\text{feedback}+\text{suppression}} \tag{1}$$

## Characterization of responses to naturalistic movie clips

Signal to noise ratio (SNR) was calculated according to *Baden et al., 2016* by

$$\text{SNR} = \frac{Var[\langle C_r \rangle]_t}{\langle Var[C]_t \rangle_r} \tag{2}$$

where $C$ is the $T$ by $R$ response matrix (time samples by stimulus repetitions) and $\langle \rangle_x$ and $\text{Var}[]_x$ denote the mean and variance across the indicated dimension, respectively. If all trials were identical such that the mean response was a perfect representative of the response, SNR would equal 1.

The sparseness $S$ of a PSTH was calculated according to *Vinje and Gallant, 2000* by

$$S = \left(1 - \frac{\left(\sum\limits_{i=1}^{n} r_i/n\right)^2}{\sum\limits_{i=1}^{n} r_i^2/n}\right)\left(\frac{1}{1-1/n}\right) \tag{3}$$

where $r_i \geq 0$ is the signal value in the $i^{th}$ time bin, and $n$ is the number of time bins. Sparseness ranges from 0 to 1, with 0 corresponding to a uniform signal, and 1 corresponding to a signal with all of its energy in a single time bin.

Response reliability was quantified according to *Goard and Dan, 2009* as the mean pairwise correlation of all trial pairs of a unit's single-trial responses. Single-trial responses were computed by counting spikes in 20ms, overlapping time bins at 1ms resolution. Pearson's correlation was calculated between all possible pairs of trials, and then averaged across trials per condition.

To detect response peaks in trial raster plots and measure their widths, clustering of spike times collapsed across trials was performed using the gradient ascent clustering (GAC) algorithm (*Swindale and Spacek, 2014*), with a characteristic neighborhood size of 20ms. Spike time clusters containing less than 5 spikes were discarded. The center of each detected cluster of spike times was matched to the nearest peak in the PSTH. A threshold of $\theta = b + 3$ Hz was applied to the matching PSTH peak, where $b = 2\,\mathrm{median}(x)$ is the baseline of each PSTH $x$. Peaks in the PSTH that fell below $\theta$ were discarded, and all others were kept as valid peaks. Peak widths were measured as the temporal separation of the middle 68% (16th to 84th percentile) of spike times within each cluster.

To determine whether V1 suppression changes dLGN responses in a divisive or subtractive manner, we fit a threshold-linear model using repeated random subsampling cross-validation. To this end, we first selected a random set of 50% of the trials for each condition for fitting to the timepoint-by-timepoint responses a threshold linear model given by $R_{supp} = s\,R_{fb} + b$, where $R_{supp} > 0$, with $s$ representing the slope and $b$ the offset. Fitting was done using non-linear least squares (scipy.optimize.curve_fit). Throughout *Figure 2*, we report the resulting $x$-intercept as the threshold. We evaluated goodness of fit ($R^2$) for the other 50% of trials not used for fitting. We repeated this procedure 1000 times and considered threshold and slope as significant if the central 95% of their distribution did not include 0 and 1, respectively.

## Characterization of responses to drifting gratings

For display of spike rasters (*Figure 3*), trials were sorted by condition. We computed orientation tuning curves by fitting a sum of two Gaussians of the same width with peaks 180° apart:

$$R(\theta) = R_0 + R_p e^{-\frac{(\theta - \theta_p)^2}{2\sigma^2}} + R_n e^{-\frac{(\theta - \theta_p + 180)^2}{2\sigma^2}} \tag{4}$$

In this expression, $\theta$ is stimulus orientation (0–360°). The function has five parameters: preferred orientation $\theta_p$, tuning width $\sigma$, baseline response (offset independent of orientation) $R_0$, response at the preferred orientation $R_p$, and response at the null orientation $R_n$.

Orientation selectivity was quantified according to *Bonhoeffer et al., 1995*; *Olsen et al., 2012* as

$$\mathrm{OSI} = \frac{\sqrt{(\sum R_k \sin(2\theta_k))^2 + (\sum R_k \cos(2\theta_k))^2}}{\sum R_k} \tag{5}$$

where $R_k$ is the response to the $k$ th direction given by $\theta_k$. We determined OSI for each unit during both feedback and suppression conditions.

We computed the first harmonic of the response $R$ from the spike trains according to *Carandini et al., 1997* to obtain the amplitude and phase of the best-fitting sinusoid, which has the same temporal frequency as the stimulus. For each trial, we calculated

$$R = (1/D) \sum_k \cos(2\pi f t_k) + i \sin(2\pi f t_k) \tag{6}$$

where $D$ is the stimulus duration, $f$ is the temporal frequency of the stimulus, and the $t_k$ are the times of the individual spikes. We excluded the first cycle to avoid contamination by the onset

response. For (*Figure 3g*), we calculated average amplitude $F_1$ by obtaining the absolute value of the complex number $R$ on each trial, before averaging across trials, to avoid potential confounds due to differences in response phase across conditions. For the comparison of response phase, we focused on the orientation which elicited the maximal cycle average response across both feedback and suppression conditions.

## Cell typing

Units were classified as suppressed by contrast (SbC) or not suppressed by contrast (non-SbC) by comparing their mean firing rates during full-screen drifting grating presentation to their mean firing rates during blank-screen presentation. Units were classified as SbC if they were visually responsive to gratings (see below) and had a median z-scored response across orientation conditions of $\leq -3$ during at least one grating experiment. Otherwise, units were classified as non-SbC. SbC units seem to constitute a sizeable fraction in our dataset, which is similar to our previous results (*Román Rosón et al., 2019*), where SbC was also found to be among the overrepresented retinal ganglion cell (RGC) types providing input to dLGN.

To identify electrode channels within the dLGN, and their relative depth, which could be useful to distinguish between shell and core, we concentrated on the RF progression as assessed with MUAe maps that were constructed using sparse noise experiments. Because RF progression is mainly along elevation, amplitudes of MUAe for each channel were collapsed across azimuth and then range normalized. Channels with normalized amplitudes higher than an empirically set threshold (0.4) were considered part of dLGN. Non-detected channels located between detected channels were also included.

Direction selectivity index (DSI, *Niell and Stryker, 2008*) was calculated for each unit as

$$\mathrm{DSI} = \frac{R_p - R_n}{R_p + R_n + 2R_0} \tag{7}$$

where $R_p$ and $R_n$ are the firing rates in the preferred and null directions, respectively, extracted from tuning curves fit to drifting grating responses (see above), and $R_0$ is baseline firing rate independent of orientation.

The RF distance from the center of the screen was calculated for each unit by finding the position of the MUAe RF for the channel on which the unit's mean spike waveform had the largest amplitude.

## Exclusion criteria

Neurons with mean evoked firing rates < 0.01 spikes/s were excluded from further analysis. For movie clips, only neurons with SNR $\geq 0.015$ in at least one of the conditions in an experiment were considered. Of this population, 2 neurons were excluded from the analysis of the parameters returned by the threshold linear model, because their $R^2$ was $lt_0$. For gratings, we converted firing rates in response to each orientation to z-scores relative to responses to the mean luminance gray screen. We only considered visually responsive neurons, with an absolute z-scored response $\geq 2.5$ to at least 1 orientation. For the analysis of response phase, we only considered neurons with a peak of the cycle average response of at least 10 Hz in both feedback and suppression conditions, and an $F_1/F_0$ ratio of at least 0.25.

## Locomotion

We used the Euclidean norm of three perpendicular components of ball velocity (roll, pitch, and yaw) to compute animal running speed. For the analysis of neural responses as a function of behavioral state, locomotion trials were defined as those for which speed exceeded 1 cm/s for at least 50% of the stimulus presentation, and stationary trials as those for which speed fell below 0.25 cm/s for at least 50% of the stimulus presentation. To quantify the effect of running vs. sitting on various response properties, the run modulation index (RMI) was defined as

$$\mathrm{RMI} = \frac{\mathrm{running} - \mathrm{sitting}}{\mathrm{running} + \mathrm{sitting}} \tag{8}$$

## Eye tracking

The stimulus viewing eye was filmed using an infrared camera under infrared LED illumination. Pupil position was extracted from the videos using a custom, semi-automated algorithm. Briefly, each video

frame was equalized using an adaptive bi-histogram equalization procedure, and then smoothed using median and bilateral filters. The center of the pupil was detected by taking the darkest point in a convolution of the filtered image with a black square. Next, the peaks of the image gradient along lines extending radially from the center point were used to define the pupil contour. Lastly, an ellipse was fit to the contour, and the center of this ellipse was taken as the position of the pupil. A similar procedure was used to extract the position of the corneal reflection (CR) of the LED illumination. Eye blinks were automatically detected and the immediately adjacent data points were excluded. Adjustable algorithm parameters were set manually for each experiment. Output pupil position time-courses were lightly smoothed, and unreliable segments were automatically removed according to a priori criteria. Finally, the CR position was subtracted from the pupil position to eliminate translational eye movements, and pupil displacement in degrees relative to the baseline (median) position was determined by

$$\theta = 2 \frac{\arcsin(d/2)}{r} \tag{9}$$

where $d$ is the distance between the pupil and the baseline position, and $r = 1.25$ mm is the radius of the eye (*Remtulla and Hallett, 1985*). Angular displacement was computed separately for $x$ and $y$ directions.

Eye position standard deviation was computed by first taking the standard deviation of the horizontal eye position at each time point across trials, and then averaging over the 5 s during which the visual stimulus was presented. We focused on horizontal eye position because horizontal and vertical eye movements tend to occur in tandem under head-fixed conditions, and the horizontal position variance is larger (*Sakatani and Isa, 2007*), thus serving as a better proxy for variance in 2D. For each experiment, trials were sorted either by the presence of optogenetic suppression of CT feedback (*Figure 1—figure supplement 2h*), or by the behavioral state of the animal as described above (*Figure 5—figure supplement 1h*). The eye position standard deviation FMI and RMI (*Figure 1—figure supplement 2i* and *Figure 5—figure supplement 1i*) were calculated in the same manner as for the neural response properties.

**Table 1.** Breakdown of sample sizes (N) for the analyses of neural data.
See text for details.

| | Neurons | Mice |
|---|---|---|
| *Figure 1f–i* | 65 | 6 |
| *Figure 2e–i* | 63 | 6 |
| *Figure 3c–e and g* | 44 | 4 |
| *Figure 3f* | 28 | 4 |
| *Figure 3h–i* | 35 | 3 |
| *Figure 4a–b* | 39 | 4 |
| *Figure 5c–f,i–l* | 66 | 6 |
| *Figure 6*, $a_{1-3}$ | 64 | 6 |
| *Figure 6*, $a_2$ | 58 | 6 |
| *Figure 6*, $a_4$ | 63 | 6 |
| *Figure 6*, $b_1$ and $b_3$ | 63 | 6 |
| *Figure 6*, $b_2$ | 58 | 6 |
| *Figure 6*, $b_4$ | 62 | 6 |
| *Figure 6*, $C_{1,3 \text{ and } 4}$ | 59 | 6 |
| *Figure 6*, $c_2$ | 56 | 6 |
| *Figure 1—figure supplement 2a* | 65 | 6 |
| *Figure 1—figure supplement 2b,g* | 57 | 6 |
| *Figure 1—figure supplement 2c* | 63 | 6 |
| *Figure 1—figure supplement 2d-f, i* | 64 | 6 |
| *Figure 1—figure supplement 2h* | | 6 |
| *Figure 1—figure supplement 3a,c* | 39 | 4 |
| *Figure 1—figure supplement 3b,j* | 63 | 6 |
| *Figure 1—figure supplement 3d* | 54 | 6 |
| *Figure 1—figure supplement 3e* | 64 | 6 |
| *Figure 1—figure supplement 3f, h* | 38 | 4 |
| *Figure 1—figure supplement 3g* | 62 | 6 |
| *Figure 1—figure supplement 3i* | 53 | 6 |
| *Figure 1—figure supplement 4e-h* | 62 | 3 |
| *Figure 1—figure supplement 4l-n* | 73 | 3 |
| *Figure 1—figure supplement 5c,d,h,i* | 19 | 1 |
| *Figure 1—figure supplement 6c-f* | 35 | 5 |
| *Figure 1—figure supplement 6g* | 65 | 6 |
| *Figure 1—figure supplement 6h* | 56 | 3 |
| *Figure 1—figure supplement 6i* | 64 | 6 |
| *Figure 1—figure supplement 6j* | 54 | 3 |
| *Figure 3—figure supplement 1a,c,e* | 44 | 4 |
| *Figure 3—figure supplement 1b,f,h,i* | 42 | 4 |
| *Figure 3—figure supplement 1d* | 36 | 4 |
| *Figure 3—figure supplement 1g* | 40 | 4 |

*Table 1 continued on next page*

*Table 1 continued*

| | Neurons | Mice |
|---|---|---|
| *Figure 3—figure supplement 1i* | 35 | 4 |
| *Figure 4—figure supplement 1a* | 42 | 4 |
| *Figure 4—figure supplement 1b,k,i* | 43 | 4 |
| *Figure 4—figure supplement 1c-d,g,i* | 65 | 6 |
| *Figure 4—figure supplement 1e* | 36 | 3 |
| *Figure 4—figure supplement 1f* | 29 | 3 |
| *Figure 4—figure supplement 1h, i* | 44 | 4 |
| *Figure 5—figure supplement 1a* | 66 | 6 |
| *Figure 5—figure supplement 1g* | 56 | 6 |
| *Figure 5—figure supplement 1c* | 57 | 6 |
| *Figure 5—figure supplement 1d-f, i* | 65 | 6 |
| *Figure 5—figure supplement 1h* | | 6 |
| *Figure 5—figure supplement 2d-g* | 57 | 6 |
| *Figure 6—figure supplement 1,$a_1$,$b_1$,$c_1$* | 37 | 4 |
| *Figure 6—figure supplement 1, $a_2$,$c_2$* | 34 | 3 |
| *Figure 6—figure supplement 1, $b_2$* | 33 | 3 |

## Analysis of pupil dilation during movies

Following (*Reimer et al., 2014*), changes in pupil area collected during movie clip presentation (e.g. *Figure 5—figure supplement 2a*) were measured at 20ms resolution. Spiking responses were binned to match the temporal resolution of the pupil change signal, masked to exclude periods of locomotion (> 0.25 cm/s), and then further masked to only include bins corresponding to the top or bottom quartiles (dilation or constriction) of the pupil area dynamics. Neural responses (firing rate, reliability, and SNR) were then calculated separately for the remaining unmasked top or bottom pupil quartile bins. To make our analyses comparable to those obtained for V1 by *Reimer et al., 2014*, we considered pupil-related response modulations as a function of instantaneous firing rate. For *Figure 5—figure supplement 2c*, we therefore separated each time point of the PSTH, determined without taking pupil size into account, into firing rate quartiles. We then computed, for each neuron, the % change in median firing rates between top and bottom pupil quartiles in each of the four firing rate quartiles. While *Reimer et al., 2014* observed a multiplicative effect of pupil size change on V1 responses to movies, our results for dLGN rather resemble an inverted U-shape pattern.

## Statistical methods

To assess statistical significance, we fitted and examined multilevel linear models (*Gelman and Hill, 2007*). Such models take into account the hierarchical structure present in our data (i.e. neurons nested in experiments, experiments nested in recording sessions, recordings sessions nested in animals), and eliminate the detrimental effect of structural dependencies on the likelihood of Type I errors (false positive reports) (*Aarts et al., 2014*). By considering the nested structure of the data, multilevel models also eliminate the need for 'pre-selecting' data sets, such as one out of several experiments repeatedly performed on the same neurons. Whenever we have several experiments per neuron, we include all of them, and also show them in the scatter plots ('observations'). We provide the sample size for each analysis in *Table 1*. To account for repeated measurements, we fitted by-neuron random intercepts and random slopes over measurement conditions (V1 control vs V1 suppressed). By-neuron random intercepts model, the difference between neurons in overall firing rates, while by-neuron random slopes model between-neuron differences in how they responded to V1 suppression. Where possible, we included random intercepts for experiments nested in recording sessions, nested in mice, and random intercepts and slopes for neurons partially crossed in experiments. In cases where the model structure was too complex for a given data set (i.e. did not converge, or gave singular fits), we simplified the random effects structure by removing one or more terms. We fit these models in R (*R Core Team, 2017*), using the *lme4* package (*Bates et al., 2015*). We estimated F-values, their degrees of freedom, and the corresponding p-values using the Satterthwaite approximation (*Luke, 2017*) implemented by the *lmertest* package (*Kuznetsova et al., 2017*). For each analysis, we provide the exact model specification and the complete output of the model (see *Data and code availability*).

Throughout the manuscript, uncertainty in estimated regression slopes is represented as $slope \pm x$, where $x$ is $2\times$ the estimated standard error of the slope.

## Acknowledgements

This research was supported by the German Research Foundation (DFG) SFB 870 TP 19, project number 118803580 (LB), DFG BU 1808/5-1 (LB), DFG SFB 1233, Robust Vision: Inference Principles and Neural Mechanisms, TP 13, project number: 276693517 (LB), and by an add-on fellowship of the Joachim Herz Stiftung (GB). We thank D Metzler for discussions regarding the multi-level modeling, M Sotgia for lab management and support with animal handling and histology, S Schörnich for IT support, and B Grothe for providing excellent research infrastructure.

## Additional information

### Funding

| Funder | Grant reference number | Author |
|---|---|---|
| Deutsche Forschungsgemeinschaft | SFB 1233, Robust Vision: Inference Principles and Neural Mechanisms, TP 13, project number: 276693517 | Laura Busse |
| Deutsche Forschungsgemeinschaft | SFB 870, TP 19, project number: 118803580 | Laura Busse |
| Deutsche Forschungsgemeinschaft | DFG BU 1808/5-1 | Laura Busse |
| Joachim Herz Stiftung | add-on fellowship | Gregory Born |

The funders had no role in study design, data collection and interpretation, or the decision to submit the work for publication.

### Author contributions

Martin A Spacek, Conceptualization, Data curation, Investigation, Methodology, Software, Visualization, Writing – review and editing; Davide Crombie, Data curation, Methodology, Software, Writing – review and editing; Yannik Bauer, Investigation, Software, Visualization; Gregory Born, Data curation, Software, Visualization, Writing - original draft, Writing – review and editing; Xinyu Liu, Investigation, Software; Steffen Katzner, Data curation, Formal analysis, Software, Visualization, Writing – review and editing; Laura Busse, Conceptualization, Data curation, Funding acquisition, Project administration, Supervision, Writing - original draft, Writing – review and editing

### Author ORCIDs

Martin A Spacek http://orcid.org/0000-0002-9519-3284
Yannik Bauer http://orcid.org/0000-0003-2613-6443
Gregory Born http://orcid.org/0000-0003-0430-3052
Steffen Katzner http://orcid.org/0000-0002-4424-2197
Laura Busse http://orcid.org/0000-0002-6127-7754

### Ethics

All procedures complied with the European Communities Council Directive 2010/63/EU and the German Law for Protection of Animals, and were approved by local authorities, following appropriate ethics review.

### Decision letter and Author response

Decision letter https://doi.org/10.7554/eLife.70469.sa1
Author response https://doi.org/10.7554/eLife.70469.sa2

## Additional files

### Supplementary files

• Transparent reporting form

## Data availability

Data and source code used to generate the figures in the manuscript has been made available on G-Node (https://doi.org/10.12751/g-node.58bc8k).

The following dataset was generated:

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
