## [Editor Report]

This paper will be of interest to neuroscientists interested in understanding the role of corticothalamic feedback in coding of sensory inputs. The authors show that feedback is stronger for natural stimuli compared to artificial stimuli. Surprisingly, the feedback from the cortex works in parallel with other modulatory influences reflecting changes in the arousal (measured here with pupil size) or changes in locomotion.

---

## [Decision Letter]

**Decision letter after peer review:**

Thank you for submitting your article "Robust effects of corticothalamic feedback during naturalistic visual stimulation" for consideration by *eLife*. Your article has been reviewed by 2 peer reviewers, and the evaluation has been overseen by a Reviewing Editor and Tirin Moore as the Senior Editor. The reviewers have opted to remain anonymous.

The authors present important data for the role of corticothalamic feedback in awake behaving animals. They find that cortical feedback serves as a gating mechanism for the transmission of signals from retina to cortex, specifically for natural scenes. Both Reviewers supported publication provided the manuscript is edited to improve clarity.

*Reviewer 1:*

Variability in pupil dilation can be examined even for repeated natural video presentations as in Figure 5 of Reimer et al. 2014.

The visual presentation could be improved by a few small changes – this may seem nitpicky but I think it would make it easier to follow the argument from the figures:

1. Using a more distinct color (i.e. not gray) for the feedback suppression period throughout with a legend for "Stimulus On" and "Light On" for these bars in each figure.

2. Using more distinct (and potentially colorblind-friendly) colors for feedback vs suppression (rather than red and pink which were hard to distinguish even with normal color vision).

3. Rather than "Feedback" and "Suppression" something like "V1 Control" and "V1 inhibited" or "Feedback Inhibited" would be helpful.

4. In most cases there seems to be plenty of room for each scatter plot to have it's own y axis label.

5. Wherever there is room, having legends directly on the figures would be helpful, for example 3a3 and 3i there's no reason to make the reader go back and forth to the caption.

6. There are a lot of scatter plots, and including R^2^ and p values directly on the figures would be helpful, or at least indicating which ones are significant.

Would be good to indicate directly in the main text whether trials with running in figure 5 are completely overlapping with running or else what the criteria is.

Figure 3 has neurons 1 and 3 but no neuron 2.

In Figure 4a, some of the lines don't connect with dots on either side.

Was there any difference between shell and core effects? If this was mentioned I missed it.

Does silencing V1 cause the pupil to dilate? This is not essential to the paper, but the authors have the data in hand and I've wondered if becoming suddenly and temporarily cortically blind might drive a startle response, especially in earlier sessions.

*Reviewer 2:*

Spacek et al. study the corticothalamic feedback of different visual stimuli on visual thalamus. With optogenetic suppression of visual cortex feedback and simultaneous multi-channel recordings in visual thalamus, the authors succeeded to acquire important data about this essential feedback loop in awake, behaving animals. The paper is technically very impressive and the results are important for a wide range of readers.

My concerns and suggestions:

The Abstract is not mentioning any methods and therefore the reader has no clue where the results come from. Please change.

The long Discussion should be structured by introducing headlines.

The Introduction, Result, and Discussion is not up to date with the current literature. There were several papers published during the last 2 years which are not or only barely mentioned, but which are relevant for this manuscript. Please adapt.

For example, several imaging papers of layer 6 were lately published (e.g. Liang et al. 2021, Augustinaite et al. 2020). So, layer 6 is not as mysterious any more as claimed in the Introduction and Discussion.

The physiology of dLGN TC neurons should be better described in the Introduction. The authors mention mGluR, but NMDA R and AMPA R are as important or even more important for the interpretation of the results. Especially, NMDA plateau potentials in the dendrites of TC neurons can easily explain the observed gain of CT feedback (Augustinaite et al. 2014). Therefore, the statement in line 455-456 is not necessary – it can be easily replaced by a few sentences about TC neuron physiology.

dLGN receives modulatory feedback from visual cortex L6 and modulatory input from brainstem. In this light, it is not surprising that locomotion and CT feedback are to some extent independent. This should be mentioned in the Discussion and the Abstract.

The resting state has a wide range of attentive states which can be determined by the pupil diameter (see for example J. Reimer, M.J. McGinley, Y. Liu, C. Rodenkirch, Q. Wang, D.A. McCormick, A.S. Tolias; Pupil fluctuations track rapid changes in adrenergic and cholinergic activity in cortex. Nat. Commun., 7 (2016), p. 13289). The state of the animal is specifically important due to the spike/burst controversy (see for example, Steriade, M. To burst, or rather, not to burst. Nat Neurosci 4, 671 (2001) https://doi.org/10.1038/89434). The authors should comment if the pupil diameter, and thereby the state of the animal, was in any way taken into account during the analysis, especially for the extraction of bursts. Do the authors observe changes in pupil diameter?

Related to the previous comment, the authors should discuss how reliably (calcium) busts can be separated from high frequency firing.

When the video presentation ends, neuron #1 in Figure 1d shows a strong off response with the cortical inhibition still on. Is this a typical behavior? Can the authors explain?

---

## [Author Response]

Reviewer 1:Variability in pupil dilation can be examined even for repeated natural video presentations as in Figure 5 of Reimer et al. 2014.

We thank the reviewer for this remark. We have implemented the method proposed by Reimer et al. (2014) and performed the analyses for our dLGN data, comparing ring rates in the absence of locomotion, for more rapidly than usual dilating vs. constricting pupil sizes. We have done these analyses separately for trials in control conditions vs. V1 suppressed. We consistently observed that pupil dilation is associated with enhanced ring rates, and this effect persists even if CT feedback is suppressed. These new results strengthen the conclusions that were based on using locomotion as a marker of behavioral state.

The visual presentation could be improved by a few small changes – this may seem nitpicky but I think it would make it easier to follow the argument from the figures:1. Using a more distinct color (i.e. not gray) for the feedback suppression period throughout with a legend for "Stimulus On" and "Light On" for these bars in each figure2. Using more distinct (and potentially colorblind-friendly) colors for feedback vs suppression (rather than red and pink which were hard to distinguish even with normal color vision).

We thank the reviewer for helping us to improve our visualization. We have revised the color palette for our plots and added the legends.

3. Rather than "Feedback" and "Suppression" something like "V1 Control" and "V1 inhibited" or "Feedback Inhibited" would be helpful.

Thank you for the suggestion. We followed the reviewer’s advice and now label the axes with V1 Control and V1 Suppressed.

4. In most cases there seems to be plenty of room for each scatter plot to have it's own y axis label

We have added y axis labels in all places where we felt that they would be helpful. In other places, we preferred to share y axis labels across panels to reduce unnecessary clutter.

5. Wherever there is room, having legends directly on the figures would be helpful, for example 3a3 and 3i there's no reason to make the reader go back and forth to the caption.

Agreed. We have added more fi gure legends.

6. There are a lot of scatter plots, and including R^2^ and p values directly on the figures would be helpful, or at least indicating which ones are significant.

We have now added p values directly to the figures.

Would be good to indicate directly in the main text whether trials with running in figure 5 are completely overlapping with running or else what the criteria is.

We have added the requested information to the main text, as suggested by the reviewer.

Figure 3 has neurons 1 and 3 but no neuron 2.

Our intent was to use the same labels for the example neurons throughout the text. For Figure 3, we want to introduce another example neuron (i.e., neuron 3, which we had not shown yet in Figure 1 and Figure 2), because this neuron compellingly exempli es that effects of CT feedback can also enhance responses to gratings. We have clarified this in the respective figure captions.

In Figure 4a, some of the lines don't connect with dots on either side.

We apologize for the confusion. We randomly jittered the horizontal position of the points to avoid overlap, but the lines connecting the pairs still end at the central position to represent the actual values. To avoid future misunderstanding, we have added a brief explanation to the figure caption.

Was there any difference between shell and core effects? If this was mentioned I missed it.

We thank the reviewer for the interesting question. We analyzed the effects of V1 suppression more generally as a function of depth relative to the top-most channel recorded in dLGN, in order to avoid having to binarize into putative shell and core. These results are shown in panels Figure 1S3b and g, and Figure 3S1b and g. To increase the clarity of presentation, we have now stated in the methods and in the caption of Figure 1S3 that depth could serve as a proxy for dLGN shell and core.

Does silencing V1 cause the pupil to dilate? This is not essential to the paper, but the authors have the data in hand and I've wondered if becoming suddenly and temporarily cortically blind might drive a startle response, especially in earlier sessions.

We thank the reviewer for the interesting question. We have now examined how pupil size changes during V1 suppression (Figure 1-Supplement 6). Contrary to the hypothesis raised by the reviewer, we find either no change at all (in the case of direct L6CT suppression) or a slight pupil constriction (in the case of suppression via PV activation). We believe that the slight pupil constriction in the latter case is indicative of some light leaking through the shield that is attached to the well around the craniotomy, or perhaps through the brain itself to the retina, rather than an effect of arousal. Indeed, given the exquisite light sensitivity of stGtACR2, we used lower light intensities for direct photosuppression in Ntsr1-cre mice (median intensity to activate stGtACR2: 10.84 mW/mm^2^) than in PV-Cre mice (median intensity to activate ChR2: 13.66 mW/mm^2^).

After performing a number of new control analyses, we are convinced that it is unlikely that the neural effects of V1 suppression are mediated through potential light leakage. First, the variation of pupil size across conditions is much larger than the small constriction that we observe with photostimulation. Second, the effects of V1 suppression on dLGN neural responses were unrelated to the amount of pupil constriction (Figure 1-Supplement 6g j). Third, we analyzed additional experiments performed in a Crenegative Ntsr1 mouse that underwent the same surgical and injection procedures as the experimental animals. We found no effects of optogenetic light stimulation (Figure 1-Supplement 5).

Overall, we would like to thank the reviewer for the detailed read and thoughtful suggestions how to improve our manuscript.

Reviewer 2:Spacek et al. study the corticothalamic feedback of different visual stimuli on visual thalamus. With optogenetic suppression of visual cortex feedback and simultaneous multi-channel recordings in visual thalamus, the authors succeeded to acquire important data about this essential feedback loop in awake, behaving animals. The paper is technically very impressive and the results are important for a wide range of readers.My concerns and suggestions:The Abstract is not mentioning any methods and therefore the reader has no clue where the results come from. Please change.

We thank the reviewer for pointing this out. We have revised the abstract and modified the title.

The long Discussion should be structured by introducing headlines.

We have revised the Discussion to be more succinct and introduced headlines, as suggested by the reviewer.

The Introduction, Result, and Discussion is not up to date with the current literature. There were several papers published during the last 2 years which are not or only barely mentioned, but which are relevant for this manuscript. Please adapt.For example, several imaging papers of layer 6 were lately published (e.g. Liang et al. 2021, Augustinaite et al. 2020). So, layer 6 is not as mysterious any more as claimed in the Introduction and Discussion.

We fully agree. We apologize for the lack of specificity and recency and have updated all parts of our manuscript with respect to the current literature. For instance, in the new version of the introduction, we now explicitly highlight the important work by Augustinaite and Kuhn (2020) and Liang et al. (2021) regarding the effects of behavioral state on V1 L6CT neurons.

The physiology of dLGN TC neurons should be better described in the Introduction. The authors mention mGluR, but NMDA R and AMPA R are as important or even more important for the interpretation of the results. Especially, NMDA plateau potentials in the dendrites of TC neurons can easily explain the observed gain of CT feedback (Augustinaite et al. 2014). Therefore, the statement in line 455-456 is not necessary – it can be easily replaced by a few sentences about TC neuron physiology.

Thank you very much for this very helpful comment and the pointer to the literature. We have modified the introduction to better introduce the cellular physiology of TC neurons and mention the NMDA plateau potentials as one potential source for depolarization associated with gain modulations in the introduction and discussion.

dLGN receives modulatory feedback from visual cortex L6 and modulatory input from brainstem. In this light, it is not surprising that locomotion and CT feedback are to some extent independent. This should be mentioned in the Discussion and the Abstract.

We thank the reviewer for this helpful recommendation. We have modi ed the abstract and added a brief note to the introduction that dLGN receives neuromodulatory inputs from the brain stem, which can also have depolarizing effects on dLGN relay cells. In addition, we have expanded this point in the discussion. Yet, we still think that our finding of rather independent effects of CT feedback and neuromodulation is interesting and relevant for a number of reasons. On the one hand, recently published results indicate that our finding is rather general, because signals in visual thalamic neurons related to whisker movements also seem to be independent from visual cortical activity (Nestvogel and McCormick, 2022). On the other hand, the emerging picture might not be as straightforward as suggested, as two recent studies report a relationship, rather than independence, between CT feedback and behavioral state modulations of dLGN neurons for spontaneous activity (MolnÆr et al., 2021) or gratings (Reinhold et al., 2021). We have re-written the discussion to incorporate these new findings.

The resting state has a wide range of attentive states which can be determined by the pupil diameter (see for example J. Reimer, M.J. McGinley, Y. Liu, C. Rodenkirch, Q. Wang, D.A. McCormick, A.S. Tolias; Pupil fluctuations track rapid changes in adrenergic and cholinergic activity in cortex. Nat. Commun., 7 (2016), p. 13289). The state of the animal is specifically important due to the spike/burst controversy (see for example, Steriade, M. To burst, or rather, not to burst. Nat Neurosci 4, 671 (2001). https://doi.org/10.1038/89434). The authors should comment if the pupil diameter, and thereby the state of the animal, was in any way taken into account during the analysis, especially for the extraction of bursts. Do the authors observe changes in pupil diameter?

We thank the reviewer for the interesting question. So far, our manuscript had only taken locomotion as a proxy for behavioral state, as locomotion typically goes along with increased pupil size (Erisken et al., 2014; McGinley et al., 2015) and increased levels of arousal (McGinley et al., 2015; Vinck et al., 2015). To also study the effects of locomotion-independent arousal, we have now applied the analysis mentioned by the reviewer: following methods originally suggested by Reimer et al. (2014), we identified periods of the video presentation without locomotion that corresponded to the upper or lower quartile of pupil size change. Similar to the results that Reimer et al. (2014) found for primary visual cortex, we observed that ring rate in dLGN was enhanced during times when the pupil was dilating faster than usual vs. when it was constricting faster than usual. Like the effects of running, the modulations by pupil-indexed arousal persisted even with V1 suppression. We present these new results in Figure 5 – Supplement 2.

Related to the previous comment, the authors should discuss how reliably (calcium) busts can be separated from high frequency firing.

We thank the reviewer for this important technical comment. We fully agree that our assessment of bursting with extracellular recordings is indirect and hence less reliable than with intracellular recordings. We would like to point out, however, that our criteria for defining bursts have been shown to be a stringent criterion for selecting only those spikes originating from calcium-induced bursts (Lu et al., 1992). Accordingly, our spike trains show features that have been known from intracellular recordings of bursts riding on top of low-threshold calcium spikes, namely the lower spontaneous activity, phase advance in response to gratings, and the higher degree of non-linearity (Lu et al., 1992). We have added a cautionary note to the discussion and methods.

When the video presentation ends, neuron #1 in Figure 1d shows a strong off response with the cortical inhibition still on. Is this a typical behavior? Can the authors explain?

Based on visual inspection, we think that this behavior is fairly typical across the population of recorded neurons. At video offset, we have both an increase in luminance as well as a reduction in contrast. One might speculate that these changes at video offset might drive larger responses with V1 suppression, because ring rates in response to the video are overall lower and thus dLGN neurons might not be as exhausted as with CT feedback intact.